



# Recommendations on benchmarks for photochemical grid model applications in China: Part I – PM$_{2.5}$ and chemical species

Ling Huang[1], Yangjun Wang[1], Hehe Zhai[1], Shuhui Xue[1], Tianyi Zhu[1], Yun Shao[1], Ziyi Liu[1], Chris Emery[2], Joshua Fu[3], Kun Zhang[1], Greg Yarwood[2], Li Li[1*]

[1]School of Environmental and Chemical Engineering, Shanghai University, Shanghai, 200444, China
[2]Ramboll, Novato, California, 95995, USA
[3]Department of Civil and Environmental Engineering, University of Tennessee, Knoxville, TN 37996, USA

*Correspondence to*: Li Li (lily@shu.edu.cn)

**Abstract**

Photochemical grid models (PGMs) are being applied more frequently to address diverse scientific and regulatory compliance associated with deteriorated air quality in China for the past decade. Solid evaluation of model performances guarantees the robustness and reliability of the baseline modelling results, so subsequent applications are built on top of it; thus, model performance evaluation (MPE) is a critical step of any PGM applications. MPE procedures and associated benchmarks have been proposed for PGM applications in the United States and Europe. However, with numerous input data needed, diverse model configurations, and evolution of the model itself, no two PGM applications are exactly the same. Therefore, those MPE benchmarks proposed based on studies outside China may not be appropriate for evaluation of the increasing number of PGM applications in China. Here we follow an established approach as published in previous literatures, to recommend statistical benchmarks for evaluation of simulated particulate matter (PM) concentrations in China. A total of 128 peer-reviewed articles published between 2006 and mid-2019 that applied one of four most frequently used PGMs in China are compiled to summarize operational model performance results. Quantile distributions of common statistical metrics are presented for total PM$_{2.5}$ and speciated components. Influences of different model configurations, including modelling regions and seasons, spatial resolution of modelling grids, temporal resolution of MPE, etc., on the range of reported statistics are discussed. Benchmarks for four frequently used evaluation metrics are provided for two tiers – "goals" and "criteria", where "goals" represent the best model performance that a model is currently expected to achieve and "criteria" represent the model performance that the majority (i.e. two thirds) of studies can meet. Our proposed benchmarks are further compared with those developed for United States and Europe. Additional recommendations for MPE practices are also given. Results from this study shall help the ever-growing modelling community in China to have a better objective assessment of how well their simulation results are compared with previous studies and to better demonstrate the credibility and robustness of their PGM applications prior to subsequent regulatory assessments.

## 1 Introduction

Along with the rapid economic development and fast urbanization in China for the past several decades, serious air pollution problems have frequently occurred in many regions of China. The infamous 2013 January severe haze pollution in Beijing and surrounding areas with record-breaking hourly concentrations of PM$_{2.5}$ (particular matter with an aerodynamic diameter less than 2.5 μm) has attracted numerous attention (e.g. Tao et al., 2014; Quan et al., 2014; M. Gao et al., 2015; etc.). Tremendous efforts have been spent to mitigate air pollution situations in China, including the "Air Pollution Prevention and Control Action Plan" in 2013 (The State Council of the People's Republic of China, 2013), "Three-year Plan on Defending the Blue Sky" in 2018 (The State Council of the People's Republic of China, 2018), "Action Plan for Comprehensive Control of Air Pollution in Autumn and Winter" (The Ministry of Ecological Environment, 2018a). Annual PM$_{2.5}$



concentrations and the number of heavy haze days have been reduced in many regions across China during the past several years (Q. Zhang et al., 2019; The Ministry of Ecological Environment, 2018b). Among these efforts, photochemical grid models (PGMs) that numerically simulate the spatial and temporal distributions of air pollutants including ozone, particulate matter (PM), air toxics, and their precursors and/or products, is a key component of linking scientific researches with

regulatory applications. With its unique capabilities and features, PGMs have been utilized for a wide range of purposes, including but not limited to understanding the underlying formation mechanisms of secondary air pollutants, evaluation of air quality impacts on public health and ecosystems, developing effective control strategies towards meeting national air quality standards, and etc.

The use of PGMs is much less constrained in the sense that there are no such "uniform" settings for PGM applications. First

and foremost, there exist different photochemical models developed by different groups. To give a few examples, the GEOS-Chem by Harvard University at global scale (http://www.geos-chem.org), the Comprehensive Air Quality Model with Extensions (CAMx) by Ramboll (Ramboll Environment and Health, 2018) and the Community Multiscale Air Quality (CMAQ) model (Foley et al., 2010) by United States (U.S.) Environmental Protection Agency (EPA) at regional scale. On top of that, a PGM application requires various inputs including time-variant meteorology, hourly and gridded emissions

inventory, initial/boundary conditions (for example, from global models, or static assumptions), and land use dataset. Model configurations include chemical mechanism, vertical diffusion scheme, planetary boundary layer scheme, numerical solver, dry deposition scheme (e.g. L. Zhang et al. 2003 vs. Wesely 1989), etc. In addition, PGMs are applied with different spatial scales (from urban to regional, super-regional and even global) over different temporal scales (from episodic to monthly, seasonal, yearly or even multi-yearly). All these variations lead to a rich compilation of PGM applications that differ from

each other in one way or more.

A critical step of all PGM applications is model performance evaluation (MPE); that is to demonstrate how well modelling results can replicate the observed magnitude as well as the spatial and temporal variations of the target pollutant. Comprehensive and solid MPE practices ensure the accuracy and reliability of modelling results of a baseline PGM simulation and therefore the subsequent applications that are built on top of it. In U.S., four tiers of MPE were proposed as

regulatory modelling guidance (EPA, 2014; see full description by Dennis et al. 2010): (1) **operational evaluation**, in which quantitative, statistical and graphical comparisons are performed based on paired modelled and observed data; (2) **dynamic evaluation**, in which "*the accuracy of the model in characterizing the sensitivity of ozone and/or PM$_{2.5}$ to changes in emissions*" is analysed; (3) **diagnostic evaluation**, in which individual physical and chemical process of the model system is evaluated based on process-oriented analysis; and (4) **probabilistic evaluation**, in which "*the level of confidence in the

model predictions is assessed through techniques such as ensemble model simulations*". In most cases, only the operational evaluation is being applied for MPE and only few applications also conducted dynamic evaluation (e.g., Foley et al., 2015). The first modelling guidance document issued by EPA provided a set of ozone MPE metrics for ozone attainment demonstration (EPA, 1991). Later, Boylan and Russell (2006) introduced the concept of "**goals**" ("*the level of accuracy that is considered to be close to the best a model can be expected to achieve*") and "**criteria**" ("*the level of accuracy that is

considered to be acceptable for modelling applications*") for model evaluation. They recommended mean fraction error (MFE, <=50% for goal and <=75% for criteria) and mean fraction bias (MFB, within 30% for goal and within 60% for criteria) as the metrics for PM species evaluation. Several years later, Simon et al. (2012) conducted a comprehensive review of operational MPE results reported in peer-reviewed journals published between 2006 and 2012 on PGM applications across North America (mostly U.S.) and presented quantile distribution of most commonly reported MPE statistics. Emery et

al. (2017) later expanded the literature compiled by Simon et al. (2012) and developed an updated set of MPE benchmarks for both ozone and PM species following the concept of "goals" and "criteria" proposed by Boylan and Russell (2006). In Europe, the Forum for Air Quality Modelling in Europe (FAIRMODE) model evaluation methodology is developed to support unified model evaluation process of air quality models used by European Union Member States (Janssen et al., 2017).



The approach of FAIRMODE also relies on various statistical indicators and diagrams based on paired modelled and observed data to offer diagnostics of model performance. Many PGM applications in China used these U.S. based benchmarks to demonstrate their model robustness (e.g. J. Hu et al., 2017; D. Chen et al. 2017; Tao et al. 2018; J. Gao et al., 2017; etc.) and no doubtfully these U.S. oriented studies provide invaluable information. Nevertheless, it should be noted that all these benchmark studies were based on PGM applications mostly for US and may not be suitable for model evaluation of PGM applications in China, given the complex interactions of various model inputs and availability of local dataset (i.e. emission inventory, speciation database). Therefore, a set of statistics and benchmarks that is specifically targeted to evaluate PGM applications in China is urgently needed but is currently missing to our knowledge.

In this study, a comprehensive review of operational model evaluations of criteria air pollutants including gaseous pollutants (e.g. $SO_2$, $NO_2$, ozone) and particulate matters (e.g. $PM_{10}$, total $PM_{2.5}$, and speciated $PM_{2.5}$) based on model evaluations results of PGM applications in China published in peer reviewed journals between 2006 and 2019 (latest journal published on July 22, 2019, Du et al. 2019) was conducted. The ultimate goal of this work is to develop and recommend a set of quantitative and objective MPE benchmarks that are suitable for PGM applications in China so that the modelling community can have an objective assessment of how well their simulation results compared with historical studies and to better demonstrate the credibility and robustness of PGM applications prior to subsequent regulatory assessments. The work done by Simon et al (2012) and Emery et al (2017) provide excellent examples of methodology and thereby was mostly adopted in this study. We divided this whole work into three parts: the first part (i.e. the current one) gives a general overview of air quality modelling studies in China compiled in this study and results for $PM_{2.5}$ and speciated components are presented; results for ozone will be discussed in the second part; results for other criteria pollutants including $PM_{10}$, $SO_2$, $NO_2$, and CO, etc. will be discussed in the last part. Same as Emery et al. (2017), our proposed benchmarks should not be considered as pass/fail tests but "*simple references to the range of recent historical performance for commonly reported statistics*" (Emery et al., 2017). Evaluation of performances of meteorological inputs for PGM application is also critical, especially for applications focused on source attribution; this will be discussed in a separate study as future work.

## 2 Methodology

### 2.1 Data compilation

Over 160 peer-reviewed articles that applied regional air quality models in China and published from 2006 to mid-2019 were first compiled in this work. These studies address diverse air quality issues over entire or certain regions of China, including quantifying source contributions during heavy haze episodes, evaluating emission control schemes, accessing impact of air pollution on health effects and crop yields, etc. Four photochemical models - CMAQ, CAMx, the Weather Research and Forecasting model coupled with Chemistry (WRF-Chem, Grell et al., 2005), and the Nested Air Quality Prediction Modelling System (NAQPMS, Z. Wang et al. 2006) are covered in this compilation. While the former three models are developed by institutes and/or companies outside China, the NAQPMS is developed by the Institute of Atmospheric Physics of Chinese Academy of Sciences and has mostly been utilized for applications in China. Similar to Simon et al. (2012), we excluded studies that did not report any MPE results or only reported MPE results in graphical form, which leads to a final set of 128 articles included in this review (see summary in Table S1). We defined ten regions as shown in Figure 1, namely Beijing-Tianjin-Hebei (BTH) region, Yangtze River Delta (YRD) region, Pearl River Delta (PRD) region, Sichuan Basin (SCB), North China Plain (NCP), Northwest, Northeast, Southeast, and Southwest (see Table S2 for provinces covered in this region).



### 2.2 Metrics evaluated

A total of 20 performance metrics was used in the 128 articles compiled in this study (see Supplemental Table S3 for a complete list of the 20 metrics). In general, these statistical metrics could be divided into two types: one is to indicate how well model captures the magnitude of observations. Examples of this type include mean bias (MB), normalized mean bias

(NMB), fractional bias (FB), etc. The other type of statistical metrics is used to indicate how the model captures the variations of observations and most commonly used metrics are "correlation coefficient" or "index of agreement".

While some of the compiled studies explicitly provide mathematical formula of the MPE metrics used in their paper, quite many did not. This causes ambiguity when a common terminology or abbreviation was used but no explicit formula is provided. For example, the term of "correlation coefficient" (or "correlative coefficient") is frequently used in many studies

but turned out to be calculated using different mathematical formula in different studies. In some studies, the "correlation coefficient" refers to the Pearson correlation coefficient (R), which indicates the strength of linear relationship between observations and predictions; while in some studies, it refers to the coefficient of determination ($R^2$) that represents the fractions of predicted variations explained by observations. In these two cases, $R^2$ value is simply the square of R value. In two studies (X. Wang et al., 2018; H. Zhang et al., 2018), the term of "correlation coefficient" is used but formulated as the

root mean square error (RMSE). To make things even more complicated, this correlation coefficient is used to indicate model's capability of capturing temporal variations in most of the studies but also spatial variations in rare cases (e.g. Ge et al., 2014). For temporal variations, this "correlation coefficient" is calculated based on temporally (hourly or daily) matched observation and modelled results at a single monitoring site (or averages across multiple monitoring sites in many cases). For spatial variations, this "correlation coefficient" is calculated based on pairs of observations and modelled results at multiple

sites and its value is used to demonstrate spatial performance. To have better comparability among studies, we converted $R^2$ values to R. "Index of Agreement" (IOA) is another example that cautions must be taken when collecting data since the definition of IOA is not unique among these studies. Most of the studies use the definition of IOA (*d*) shown in Table 1 and only one study used the formula in Table 3. The use of IOA is discussed more in section 3.4 and we dropped the second formula for developing IOA benchmarks.

### 2.3 Derivation of benchmarks

In this study, the method established by Simon et al. (2012) and Emery et al. (2017) was mostly adopted. Quartile distribution for each statistical metrics (depending on the data availability) was first presented and the influences of several model key inputs on these metrics were discussed. Rank-ordered distribution for selected metrics was then used to pick out the 33[rd] and 67[th] percentiles. According to Emery et al. (2017), the 33[rd] and 67[th] percentile separates the whole distribution

into three performance range: studies that fall within the 33[rd] percentile can be considered as successfully meeting the goals that the best a model is currently expected to achieve; studies that fall between 33[rd] and 67[th] quantiles indicate successfully meeting the criteria that the majority of studies could achieve; studies that fall outside the 67[th] quantile indicate relative poor performance for that specific metric. A summary table with values of 33[rd] and 67[th] quantile values for recommended statistical metrics is provided at the end this work and is compared with U.S. benchmarks proposed by Emery et al. (2017).

### 3. Results

### 3.1 General overview of air quality modelling studies in China

A total of 128 articles with PGM applications published between 2006 and 2019 were compiled in this work. Figure 2a shows the number of articles published in each year during the past 14 years. Prior to 2013, number of studies that utilized PGMs in China was generally limited. A noticeable increase of number of studies was apparent in 2013 with doubled or

even tripled studies each year during 2016-2019. This sharp increase coincides with the infamous record-breaking haze event





in January 2013 that attracted numerous attentions to air pollution issues in China. Since then, series of air pollution related actions were carried out due to increasing funding that became available for the research community to perform various studies related to air pollution. Of the 128 articles included in this work, WRF-Chem was the most frequently used PGM (used in 56 studies), followed by CAMx (31 studies), CMAQ (27 studies), and NAQPMS (18 studies). One study evaluated

model performances for CAMx, CMAQ, and NAQPMS (Q. Wu et al. 2012). In terms of regions, BTH (56 studies), YRD (35 studies), and PRD (25 studies) are the top three most evaluated regions (Figure 1) (note that we excluded studies that cover entire China for this count).

Meteorological data are needed to drive air quality simulations and the performance of meteorology modelling is one of uncertainties for air quality modelling performance. Meteorological data are dominantly simulated by the Weather Research

Forecasting (WRF) model (Skamarock et al., 2005) in our compiled studies or the Fifth Generation Penn State/NCAR Mesoscale Model (MM5) (Grell et al., 1994) in a few studies. Model performances of meteorological results should be also evaluated in addition to air quality simulation results. However, we do find a few studies that did not report any results with respect to their meteorological simulations. The model performances of meteorological results used to drive air quality simulations will be discussed as a future work.

Emission inventory is another critical input for PGM applications and the accuracy of emission inventory being used no doubtfully directly affects the model performance. Most frequently used emission inventory for anthropogenic sources include the MEIC developed by Tsinghua University (http://www. meicmodel.org), Regional Emission Inventory in Asia (REAS, Kurokawa et al., 2013), Intercontinental Chemical Transport Experiment-Phase B (INTEX-B) emissions (Q. Zhang et al., 2009), MIX Asian anthropogenic emissions developed by the Model Inter-Comparison Study for Asia (MICS-Asia)

emission group (M. Li et al., 2017b), and many locally developed emission inventory at regional or city-scale. For biogenic emissions, the Model of Emissions of Gases and Aerosols from Nature (MEGAN, Guenther et al., 2006) is the dominant one being used.

The national monitoring stations from the China National Environmental Monitoring Center (CNEMC) are the dominant observational data source used for model validation. The coverage of the national monitoring system increased from 74

major cities in 2013 to 338 cities across China in 2018. However, since only criteria pollutants (namely $PM_{2.5}$, $PM_{10}$, $SO_2$, $O_3$, $NO_2$ and CO) are measured at the national monitoring sites, model validation of speciated $PM_{2.5}$, ammonia, volatile organic compounds (VOCs) species (e.g. isoprene, formaldehyde), and etc. are based on measurements obtained from local monitoring sites or field observations conducted by individual research groups or institutes.

Figure 2b shows the frequency of use for each statistical metric compiled in this study. Table 2 shows the formula of metrics

that have been used in more than 10 studies. Same as Simon et al. (2012), the top three most frequently used metrics is correlative coefficient (R, 85 studies), normalized mean bias (NMB, 80 studies), and mean bias (MB, 58 studies). Other frequently used (>10 studies) metrics include root mean square error (RMSE, 54 studies), normalized mean error (NME, 50 studies), fraction bias (FB, 32 studies), index of agreement (IOA, 33 studies), fraction error (FE, 29 studies), and mean error (ME, 11 studies). Mean normalized bias (MNB) and mean normalized error (MNE) were only used in six and four studies,

respectively, as mentioned in Simon et al. (2012) that these two metrics tends to give more weight to data at low values. About 71% of articles included in this work used at least three statistical metrics for model performance evaluation (Figure 2c); 13% of studies reported numerical values for only one metric; studies included more than five MPE metrics were less than 10%; three studies (X. Li et al., 2015; Kim et al., 2017; X. Li et al., 2018) used eight statistical metrics. In terms of number of pollutants evaluated in each study (Figure 2d), 55 studies (43%) evaluated only one pollutant and 96 studies (75%)

evaluated less than or equal to three pollutants; one study (Tie et al., 2013) evaluated 12 pollutants (including several VOCs species).

Figure 3 shows the number of studies broken down by pairs of pollutants and PGM models and pairs of pollutants and metrics. As expected, $PM_{2.5}$ is the most frequently evaluated pollutant, followed by ozone, $NO_2$, $PM_{10}$ and $SO_2$, all of which





are criteria pollutants included in China's National Ambient Air Quality Standards (NAAQS). Evaluation of speciated PM species, including nitrate, sulfate, ammonium and organic carbon (OC) is about one fourth frequent as total $PM_{2.5}$ and was only covered in applications for certain regions due to limited observations.

**3.2 Quantile distributions of $PM_{2.5}$ and speciated components**

Figure 4 shows quantile distribution of eight most frequently used model performance metrics for $PM_{2.5}$ and speciated components (corresponding values are listed in Table S2). For total $PM_{2.5}$, slightly more studies reported positive MB values and negative NMB values while approximately equivalent number of studies reported both positive and negative FB values. Reported bias for $PM_{2.5}$ ranges from as low as -40 $\mu g/m^3$ to as much as 50 $\mu g/m^3$ (outliers excluded) with median values around 5 $\mu g/m^3$. The bias range for speciated components is much smaller (within 20 $\mu g/m^3$) because the absolute magnitude

of speciated components is much smaller. In terms of normalized bias, the range of $PM_{2.5}$ is comparable or smaller than speciated components. Speciated $PM_{2.5}$ tends to be dominantly under-estimated except for elemental carbon (EC), which is directly emitted from sources as opposed to other speciated components that could be both emitted directory from sources (i.e. primary) and formed via chemical reactions of precursors (i.e. secondary). Model under-estimations of secondary species (organic and inorganic) have been reported in numerous studies with explanations of missing formation mechanisms,

uncertainties with the emission inventory, and meteorology errors that were carried over, etc. For error metrics, total $PM_{2.5}$ performs better than speciated components in terms of NME, with a median NME value around 45%. For FE, median values for all PM species (except organic matters (OM)) are within 40~60%.

R and IOA are used to indicate how well the model could capture the variations of observed values and both values are within the range of 0~1. We converted $R^2$ values to R for better comparability. For total $PM_{2.5}$, median IOA value is 0.76

while median R is 0.60 ($R^2$=0.36). Minimum IOA value reported for total $PM_{2.5}$ is 0.44 while minimum R value approaches to zero. Six studies (L. Li et al., 2018; Cheng et al., 2013; L. Chen et al., 2017; X. Li et al., 2018; Y. Liu et al., 2017; Shimadera et al., 2014) reported both R and IOA values that enable inter-comparisons of the two metrics based on identical sets of data points. It is found that IOA values always tend to be higher than R values (30 out of 32 data pairs). Compared to total $PM_{2.5}$, secondary inorganic aerosols (i.e. sulfate, nitrate, and ammonium) demonstrate better performances in terms of R

values but slightly poorer performances in terms of IOA values. OM and elemental carbon (EC) show lower values for both R and IOA compared to total $PM_{2.5}$.

*Impact of season*

There are numerous factors that could affect model performances results, to give a few examples, the study region and period, source of emission inventory, model grid resolution, the temporal resolution of paired observations and modelling

results used for model evaluation, etc. We first look at NMB results of total $PM_{2.5}$ and selected species (due to availablilty of data points) by season (Figure5). For total $PM_{2.5}$, number of data points reported for fall and winter is significantly higher than those reported for spring and summer as heavy haze episdoes generally occur in fall and winter. More studies reported negative bias of total $PM_{2.5}$ for all four seasons except spring. The underestimation of total $PM_{2.5}$ in summer and fall is accompanied by dominant understimation of PM components in these two seasons (except for ammoinium in summer).

Sulfate tends to be overwhelmingly underesmiated regardless of season, which is commonly reported in literatures with potential causes of missing formation mechanisms (e.g. heterogeneous reactions, Ye et al., 2018; L. Huang et al. 2019; Shao et al., 2019). Nitrate is heavily underestimated in summer but since nitrate concentrations tend to be low under high temperature, this negative bias does not much affect the total mass of $PM_{2.5}$. In winter, nitrate is equivalently over- and under-estimated but over-estimation could be as much as 60% in terms of NMB. As opposed to sulfate and nitrate,

ammonium could be overesimated in summer. However, it should be noted that the large positive NMB values of ammonium (> 20%) in summer reported here are from one single study that was conducted at a national nature reserve in Sichuan basin (Qian et al. 2015), where prettly low ammonium concentration (< 1 $\mu g/m^3$) was observed and shall not be



considered as a representative case. OM also tends to be more underestimated, especially in summer and fall. The understimation of organic components, especially the secondary organic aerosols (SOA), is well documented by many studies (e.g. Jimenez et al., 2009; Q. Chen et al., 2017; B. Zhao et al., 2016).

*Imapct of region*

We also look at whether there are any regional differences in these statistical metrics. Constrained by number of data points, we only compared results of R and NMB for total $PM_{2.5}$ and secondary inorganic species over three key regions in China, that is the Beijing-Tianjin-Hebei (BTH) region in north China, the Yangtze River Delta (YRD) region in eastern China, and the Pearl River Delta (PRD) region in south China. These three regions represent the most populated, economically developed and urbanized city clusters in China. With respect to the total $PM_{2.5}$, R and NMB values for the three regions do

not exhibit substantial differences. More positive NMB values were reported for total $PM_{2.5}$ in YRD while the opposite trend is observed for BTH and PRD. In terms of NMB, PRD shows better performance results with smaller range of NMB (within $\pm25\%$) whereas ranges for the other two regions are within $\pm45\%$. For sulfate and ammonium, underestimation is observed for all three regions with most underestimation in YRD. For nitrate, studies in BTH and PRD reported both positive and negative NMB while nitrate in YRD is always underestimated.

*Imapct of temporal and spatial resolution*

Although PGM are usually conducted at hourly time step, validation of modelling results is not always performed with pairs of hourly data, which depends on the temporal resolution of observational data as well as the purpose of the application. Daily, weekly, monthly and even annually-averaged pairs of modelling results and observations were used for model evaluation. Figure 7 shows the quantile distribution of R, RMSE, MB, NMB and NME for $PM_{2.5}$ presented by the temporal

resolution used for model validation. Model seems to better capture observed variations when coarser temporal pairs of observations and model results are used, as indicated by higher R values as temporal resolution gets coarser. Hourly and daily results of bias metrics do not show much difference. However, NME significantly improves as temporal resolution gets coarser.

Spatial resolution is a key setup for PGM applications. For applications at local or urban scale, PGM is usually configured

with two or three nested domains that were downscaled from coarser outer domain to finer inner domain. Among the 128 articles compiled in this study, a total of 20 grid resolutions was used, ranging from as coarse as 81 km to as fine as 1 km depending on the target region and the purpose of the application. While most of the studies only performed model evaluation for one modelling domain (usually the finest domain), four studies (X. Qiao et al., 2015; L. Wang et al., 2015; X. Liu et al., 2010; S. Liu et al., 2018) calculated statistical results for multiple domains. Figure 8 shows the distribution of

three statistical metrics (R, NMB, and FB) presented by model's horizontal resolution. To remove the impact of temporal resolution, results shown in Figure 8 are only based on hourly data and results with less than five data points were excluded. In terms of R values, finer spatial resolution does not necessarily improve the correlation performance between modelling results and observations. R values at the finest grid resolution (3km) range from as low as 0.12 to as high as 0.95 while at the coarsest resolution (80km) from 0.51 to 0.76. NMB seems to be moving from underestimation to overestimation as grid

resolution gets coarser and no clear trend is observed for FB. The range of each statistical metrics seems to be more associated with the number of available points instead of the grid resolution. For example, the wider range of R and NMB at 3 km and 4 km resolution and that of FB at 12 km resolution is more likely due to more data points being available. As mentioned above, many factors could affect model performances. Thus it is difficult to solely evaluate whether there is a systematic improvement of model performances as the modelling resolution gets finer. L. Wang et al. (2015) reported results

for evaluating hourly $PM_{2.5}$ at two spatial resolutions (12 km vs. 36 km) simultaneously. For this particular study, model over-predicted $PM_{2.5}$ at 12 km resolution (positive values of MB, NMB, and FB) but under-predicted $PM_{2.5}$ at 36 km resolution (negative values of MB, NMB, and FB). This is likely due to the dilution effect that makes model results lower at 36 km domain.



### 3.3 Recommended metrics and benchmarks

We presented similar diagrams as Emery et al. (2017) to develop metrics and benchmarks for model evaluation. Figure 9 shows the rank-ordered distribution of R, IOA, NMB and NME results for total $PM_{2.5}$ and speciated components from all studies compiled in this work. Results of R for total $PM_{2.5}$ are further split into hourly (h), daily (d) and monthly (m)

resolution since it increases as temporal resolution changes from hourly to monthly. The 33[rd] percentile value increases from around 0.5 for hourly and daily to 0.70 for monthly results; the 67[th] percentile increases from 0.64 to 0.91 as the total $PM_{2.5}$ is evaluated with coarser resolution. Secondary inorganic species (sulfate, nitrate and ammonium) show consistently higher correlation coefficient compared to total $PM_{2.5}$ with relative similar range of 0.65~0.75 to above 0.80 over the 33[rd] – 67[th] percentile interval. For OC/OM, the 33[rd] (0.51) and 67[th] (0.74) percentile value is similar to that of daily $PM_{2.5}$ while EC

shows slightly lower 33[rd] (0.43) and 67[th] (0.66) percentile value compared to OC/OM. In terms of IOA, the 33[rd] – 67[th] percentile interval ranges from 0.69 to 0.91 for total $PM_{2.5}$, 0.6 to 0.83 for sulfate and nitrate, 0.73 to 0.77 for ammonium and 0.57 to 0.62 for OC/OM. Values for EC were not shown due to limited data. For bias and error, total $PM_{2.5}$ exhibits smaller values compare with speciated components, due to potential compensating effects from different components. The 33[rd] percentile NMB for total $PM_{2.5}$ is less than 10% while the 67[th] percentiles less than 20%. Among these three secondary

inorganic species, the bias and error of nitrate exhibits largest variability (NMB ranges from 16.4% to 51.0% and NME from 46.5% to 63.5% for 33[rd] to 67[th] percentile interval). The 33[rd] to 67[th] range of NMB for EC (12.0% to 39.0%) is much lower than that for OC/OM (34.7% to 59.6%) while NME for OC/OM and EC is similar, ranging from ~43% to 58%.

Based on our analysis above as well as previous conclusions from Emery et al. (2017), we propose recommended statistical metrics and associated benchmarks for total $PM_{2.5}$ and speciated component as shown in Table 2. Shaded values indicate that

less than 10 data points were available to develop the benchmarks. Values for "goal" indicate that roughly the top one third of studies could meet the benchmarks and represent the best that a model is currently expected to achieve. Values for "criteria" indicate that roughly the top two thirds of studies meet the benchmarks and represent results from the majority of studies. Our table differs from Emery et al. (2017) in three aspects. Firstly, we added benchmarks for IOA in addition to the correlation coefficient. We found a general increasing trend of using IOA for model performance evaluation since 2013

(prior to 2013, only one of our compiled studies used IOA; after 2013, 32 studies used IOA). Thus we added IOA for future reference. Secondly, we presented benchmarks for different temporal resolution of total $PM_{2.5}$ when possible. As mentioned above, R and NME results for total $PM_{2.5}$ get better as temporal resolution gets coarser while no clear trend is observed for NMB. Therefore, different benchmarks are developed for R and NME. Thirdly, Emery et al. (2017) did not present benchmarks for the correlation coefficient of speciated PM components due to large uncertainties. Here we presented

benchmarks for R and IOA of speciated PM components (except IOA for EC is not available), but cautions should be taken comparing to these benchmarks. For example, less than ten data points were used to develop the benchmarks of R for ammonium and OC/OM and IOA for ammonium. For sulfate and nitrate, although the numbers of R data points are slightly fewer than that in Emery et al. (2017), we do not observe sudden changes in the rank-order distribution as observed in Emery et al. (2017). Thus, we keep these values for future references. For NMB and NME, we do observe sharp changes in rank-

order values, for example, the NMB for nitrate and EC, and NME for EC. Therefore, we do not give benchmarks in this situation.

We further compared our results with benchmarks proposed by Emery et al. (2017). Values with an asterisk in Table 2 indicate that our benchmarks are stricter than corresponding values in Emery et al. (2017), which means results from a study would be more difficult to be considered within 33[th] (or 67[th]) percentiles if our benchmarks are used. For total $PM_{2.5}$, our

proposed benchmarks are generally stricter than that in Emery et al. (2017). For example, our NMB (NME) "criteria" value for daily $PM_{2.5}$ is 25%(45%) as opposed to 30%(50%) in Emery's study; "criteria" value for R benchmark is also higher (0.45) than those based on U.S. studies (0.40). This might partially reflect the systematic improvements in model applications (e.g. incorporation of newly discovered mechanisms) during the past several years since the latest study





included in Emery et al. (2017) was published in 2015. However, our "goal" values for NMB and R benchmarks are less strict than that proposed by Emery et al. (2017). For speciated components, NMB and NME benchmarks for nitrate and EC are lower (i.e. stricter) than Emery's study while the opposite is true for sulfate, ammonium. However, it should be noted that the numbers of data points for NMB and NME results in our study are significantly lower than that used in Emery's study, thus a direct comparison would be inappropriate. For correlation coefficient, we were only able to make a direct comparison for sulfate because of data availability and our R benchmarks for sulfate are much higher (i.e. more strict) than those in Emery's study.

### 3.4 Additional discussions and recommendations

#### *Benchmarks for European modeling community - FAIRMODE*

The air quality model benchmarking practise for PGM applications by the FAIRMODE community is somehow different from the U.S. benchmarks. The main modeling performance indicator is called the modeling quality indicator (MQI), which is calculated based on RMSE and measurement uncertainties (function of mean value and standard deviation of observations) (Janssen et al., 2017). The modeling quality objective (MQO) is the criteria value for MQI and is said to be met if MQI is less than or equal to one. In addition to the main MQI, three statistical indicators that describe certain aspects of the differences bewteen observed and modeled results – namely bias, correlation, and standard deviation are proposed as the modelling performance indicators (MPI). For each MPI, the model performance criterion (MPC) that individual MPI is expected to meet is also given. However, unlike fixed values given in this study and Emery et al. (2017), MPC is dependent on observation uncertaities. Therefore, it is not diretly comparable between MPC and the benchamrks proposed in this study or the ones in Emery et al. (2017).

#### *The use of "index of agreement"*

The concept of "index of agreement" is originally proposed by Willmott in the 1980s and has since then been widely used to "*reflect the degree to which the observed variate is accurately estimated by the simulated variate*" (Willmott, 1981) in a variety of fields. IOA has gone through several modifications (together referred as Willmott indices) since it was proposed in the original formula (Willmott 1982; Willmott et al., 1985, 2012). The formula of the original one ($d$) is shown in Table 2 (presented again in Table 3) and the other three ($d_1$, $d_1'$ and $d_r$) shown in Table 3. The first version of IOA is proposed over the correlation coefficient for its ability to "*discern differences in proportionality and/or constant additive differences between the two variables*" (Willmott, 1981) and this version is also the most widely used version in our compiled studies. Compared with $R^2$ values, the original IOA results systematically higher values (Valbuena et al., 2019) thus is being adopted in an increasing number of studies partially because it makes results appear "better". However, the original and also being the most widely used IOA is problematic in that too much weight is given to the large errors when squared (Willmott et al., 2012) and relatively high IOA values could be obtained even when a model is performing poorly (Willmott et al., 1985; Pereira et al., 2017). Newer versions as later proposed by Willmott overcome this problem by removing the squaring and are recommended over the original one (Willmott et al., 1985, 2012). Valbuena et al. (2019) suggested using $d^2$ instead of $d$, at least for estimating forest biomass based on remote sensing to facilitate comparison with studies using correlation coefficient. Over a quarter (33 studies) of our compiled studies used the "index of agreement" for MPE but only one study (Y. Peng et al. 2011) used the second formula ($d_1$) while the rest studies all used the original formula. There seems to be an increasing trend of using IOA (the original formula) as a model performance indicator for PGM applications in China (prior to 2013 only 1 study vs. 32 studies after 2013), we decided to keep IOA based results and discussions in this work for future reference but cautions should be taken when using and interpreting IOA values. It should be noted that the value of IOA alone does not necessarily tell how well the modelling results are.

#### *Additional recommendations*





Other than the recommended metrics and associated benchmarks listed in Table 2, we list additional recommendations for validation practices that would enable a complete and comprehensive picture of model performances.

(1) Provide explicit mathematical formula of statistical metrics being used to avoid any confusion. As mentioned earlier, quite many studies did not give explicit formula of used metrics in their studies. This would sometimes cause ambiguity
when a common name (for example, correlation coefficient, or index of agreement) is used but calculated using different formula.

(2) Provide as much details as possible with respect to how observation and modelling results are used to obtain the statistical results. For example, how observed data and modelled results are paired in space and time? Is any averaging performed prior to calculating statistical metrics? Specify the number of observation sites and the number of available
data points being used. This would enable a further comparison of model performances based on the amount of available data points. It should be noted that large averaging (i.e. more pairing of observed and modelled results) usually result in better statistics, but do not convey any more meaning.

(3) It is always good practise to present model performance results of meteorological fields, usually including but not limited to temperature, humidity, wind speed, and wind direction. Performance results of meteorological model could
also help explain potential causes of unsatisfactory PGM simulated results.

(4) Metrics used should always include two types of statistical metrics for model evaluation, one for magnitude evaluation (e.g. MB, NMB or FB) and one for variation evaluation (e.g. R or IOA). According to Simon et al. (2012), a minimum set of MPE statistical metrics should include "*mean observation, mean prediction, MB, ME (or RMSE) and a normalized bias and error (NMB/NME or FB/FE)*". Cautions need to be taken when presenting values of fractional
metrics, for example, NMB/NME, FB/FE. Double check if the values presented are before or after multiplied with 100%. We do find studies that present extremely small values of NMB (<1%) but should be multiplied by 100 based on the results of other evaluation metrics.

(5) Try to evaluate multiple pollutants even if the study focuses on one single pollutant. It is obvious that opposite biases in speciated PM components could compensate each other and falsely lead to a good performance of the total $PM_{2.5}$.

(6) In addition to providing numerical values of statistical metrics for model performance evaluation, graphs/plots are strongly recommended to further support model validation. To give a few examples, visualizing data via time series plots of modelled and observed data could help illustrate periods with better or poorer performances. Spatial plots with modelling results as background and observation data as dots could help demonstrate how model performs spatially.

## 4 Conclusions

With the increasing number of PGM applications in China over the past decade, a review of the model performance is needed to help understand how well these models are currently performing compared with observations and how reliable the future model applications are compared with existing studies. Following an established method used in the U.S., a total of 128 peer-reviewed studies that applied PGMs in China was compiled in this work and key information, including model applied, study region, grid resolution, evaluated metrics, and etc., were collected. As an initial attempt, operational MPE
results for total $PM_{2.5}$ and speciated components reported in the compiled studies are presented in this study; results for other pollutants and meteorological simulations will be discussed as follow-up studies. Quantile distributions of common statistical metrics used in the literature were presented and the impacts of different model configurations, including study region, study period, spatial and temporal resolutions on performance results are discussed. With the concept of "goals" and "criteria", we proposed benchmarks for four commonly used metrics – NMB, NME, R and IOA based on the method
employed by Emery et al. (2017). For total $PM_{2.5}$, we provided benchmarks with different temporal resolutions; for component species, we did not split results by temporal resolution due to limited number of data points. We kept results for index of agreement while recognizing it should be used and interpreted with cautions. Additional recommendations on good



evaluation practices are provided at the end. Results from this study could help the ever-growing modelling community in China to have a better understanding of how their model performances are compared with existing studies and also help modellers to conduct model evaluation in a more consistent fashion, which would in turn improve the comparability among different studies.

*Date availability.* All data is available upon request from the corresponding author.

*Competing interest.* The authors declare that they have no conflict of interest.

*Special issue statement.* This article is part of the special issue "Regional assessment of air pollution and climate change over East and Southeast Asia: results from MICS-Asia Phase III". It is not associated with a conference.

*Author contribution.* L.H., Y. W. and L.L. designed the research; H. Z., S. X, T. Z., and Y. S. complied studies and collected data with equal contributions; L.H. reviewed and analyzed collected data; C. E, J. F., and G. Y. provided important academic guidance; L.H. wrote the paper with contributions from all authors.

*Acknowledgement.* This study was financially sponsored by the Shanghai Sail Program (NO. 19YF1415600), the Shanghai Science and Technology Innovation  Plan (NO. 19DZ1205007), the National Natural Science Foundation of China (NO. 41875161), the Shanghai International Science and Technology Cooperation Fund (NO. 19230742500), and Chinese

National Key Technology R&D Program (NO. 2014BAC22B03 and NO. 2018YFC0213800).

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

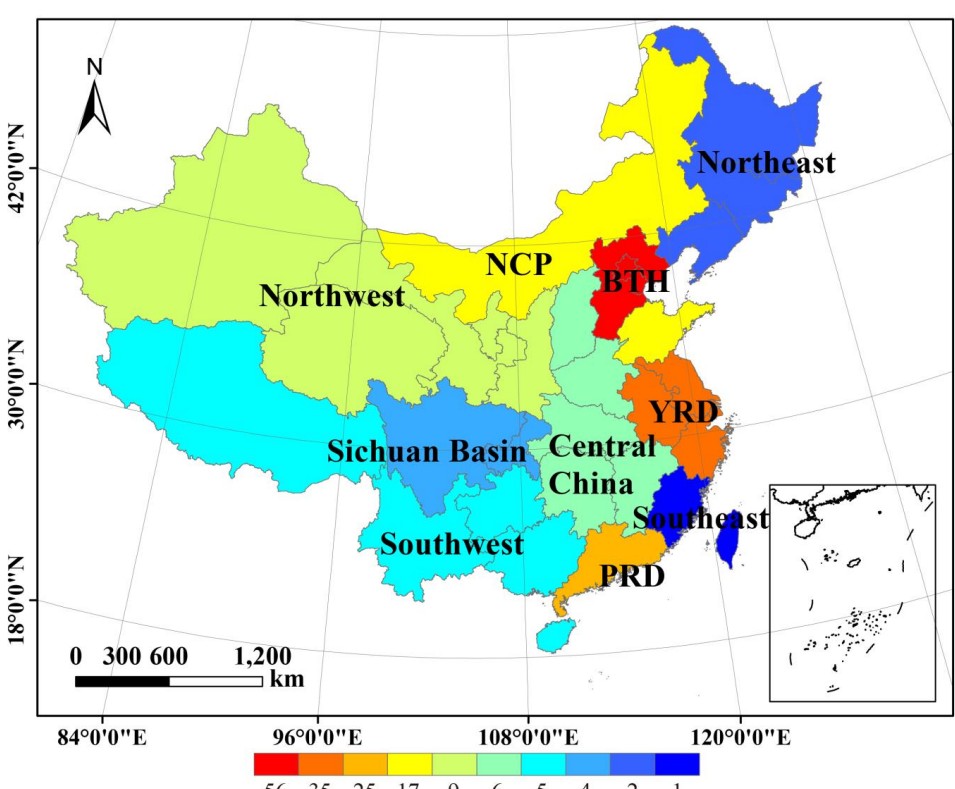

**Figure 1: Map of regions defined in this study (see Table S2 for provinces covered by each region). Colour bar indicates the number of studies evaluating the region (studies covering entire China were excluded from counting)**





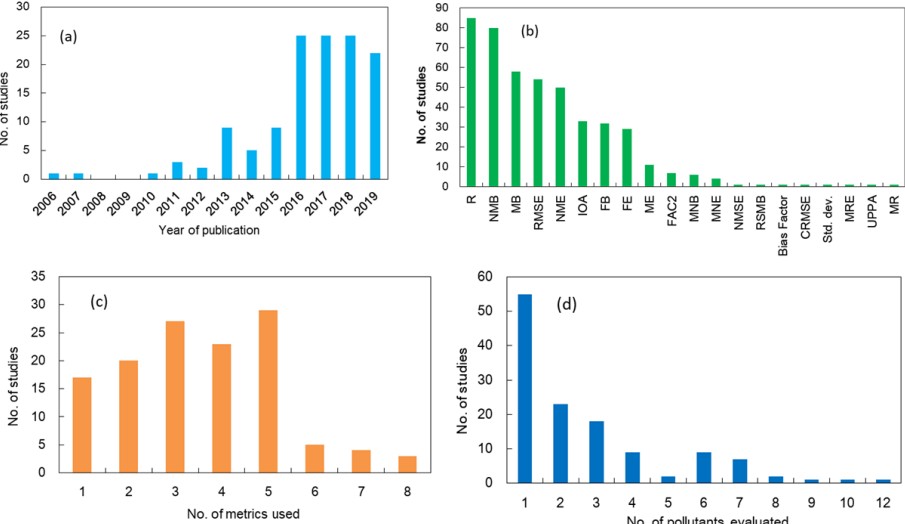

**Figure 2: (a) number of studies published during 2006-2019; (b) frequency of use of each metrics; (c) number of metrics used in studies; (d) frequency of number of pollutants evaluated.**

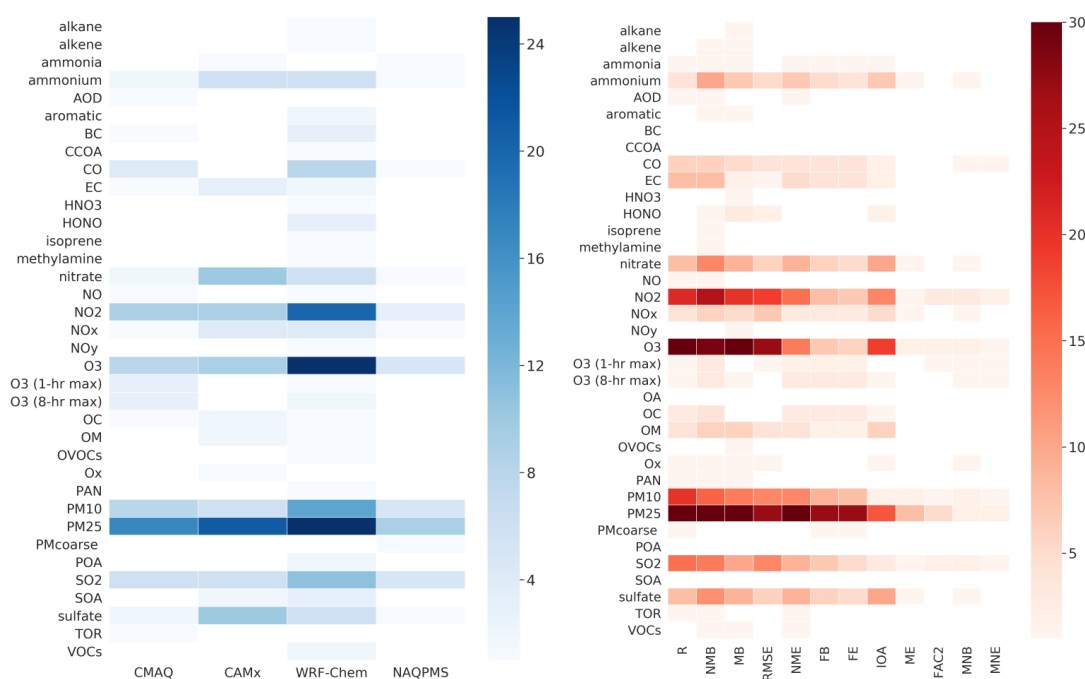

**Figure 3: Number of studies evaluating each pair of a pollutant and PGM models (left); number of studies evaluating each pair of a pollutant and statistical metric (right). See Table S4 for species abbreviations.**





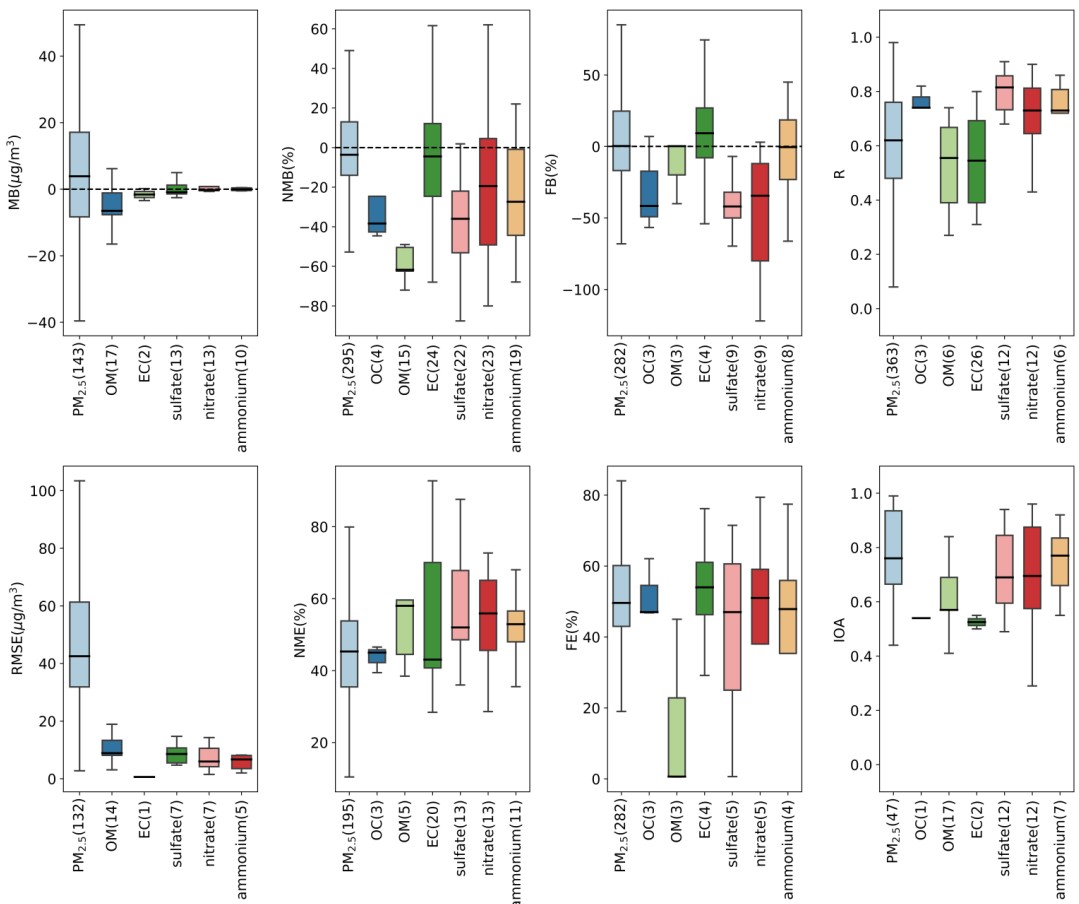

**Figure 4: Quantile distribution of selected PM performance metrics compiled in this work. Median values are shown as centerlines; the upper and lower bound of boxes correspond to the 25th and 75th percentile values; whiskers extend to 1.5 times the interquartile range (outliers are excluded).**

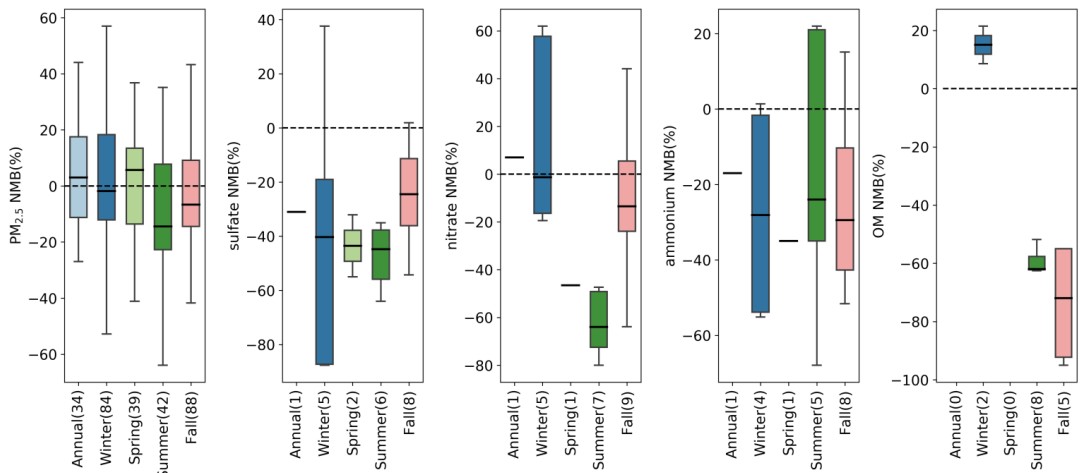

**Figure 5: NMB of total PM2.5 and speciated components split by season.**





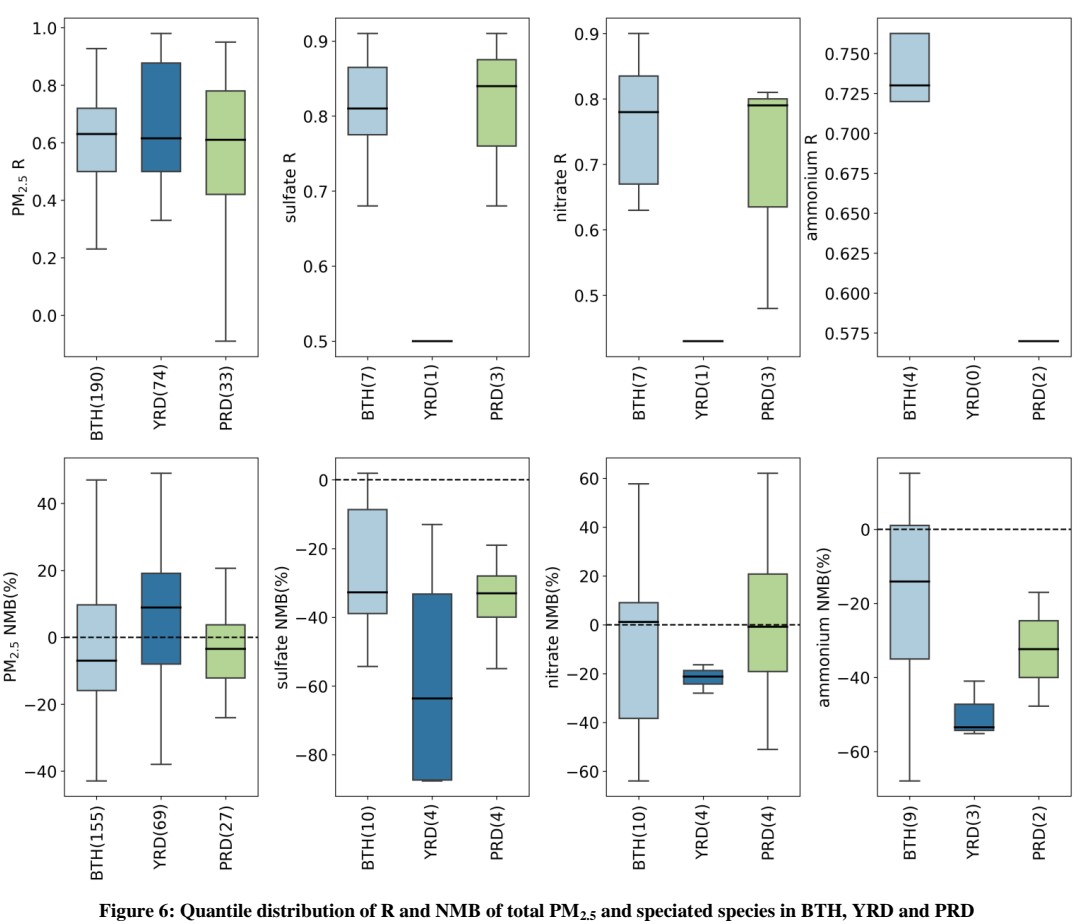

Figure 6: Quantile distribution of R and NMB of total PM$_{2.5}$ and speciated species in BTH, YRD and PRD

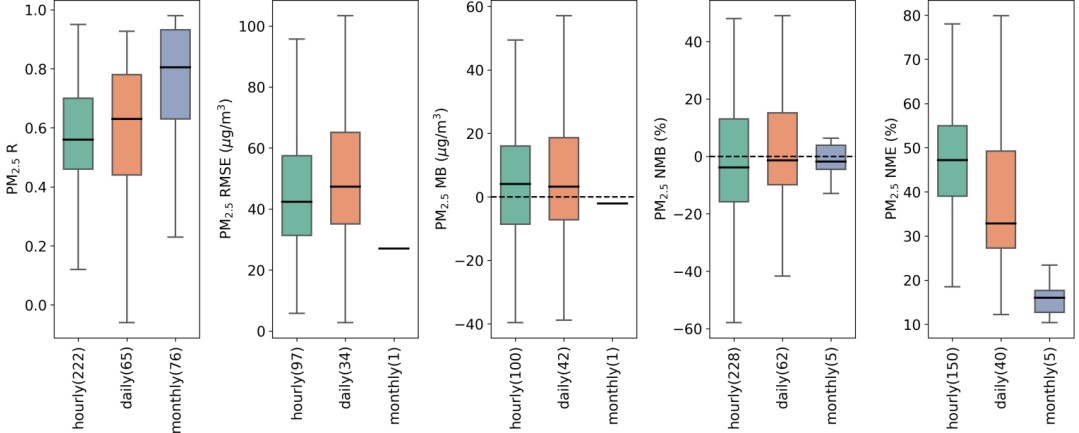

Figure 7: Quantile distributions of R, MB, NMB and NME of total PM$_{2.5}$ presented by temporal resolution for model validation




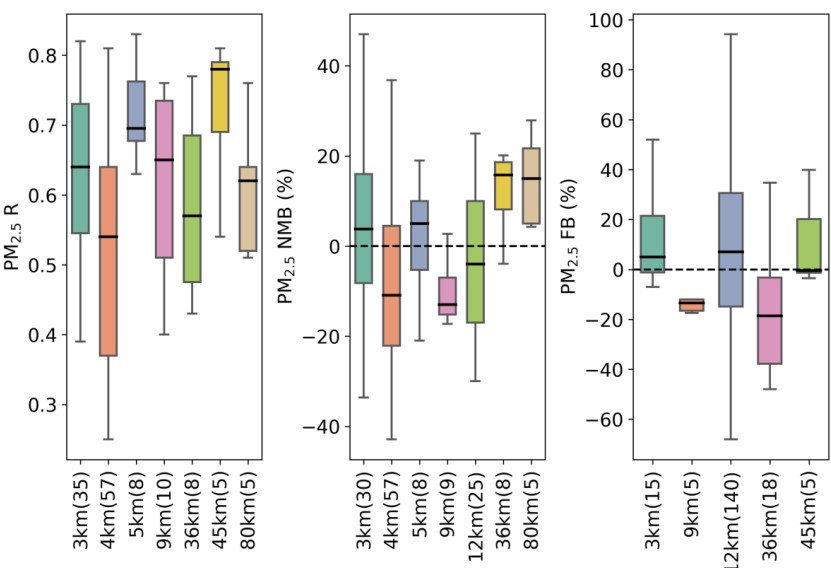

**Figure 8: Quantile distributions of R, NMB and FB of total PM$_{2.5}$ presented by model grid resolution**

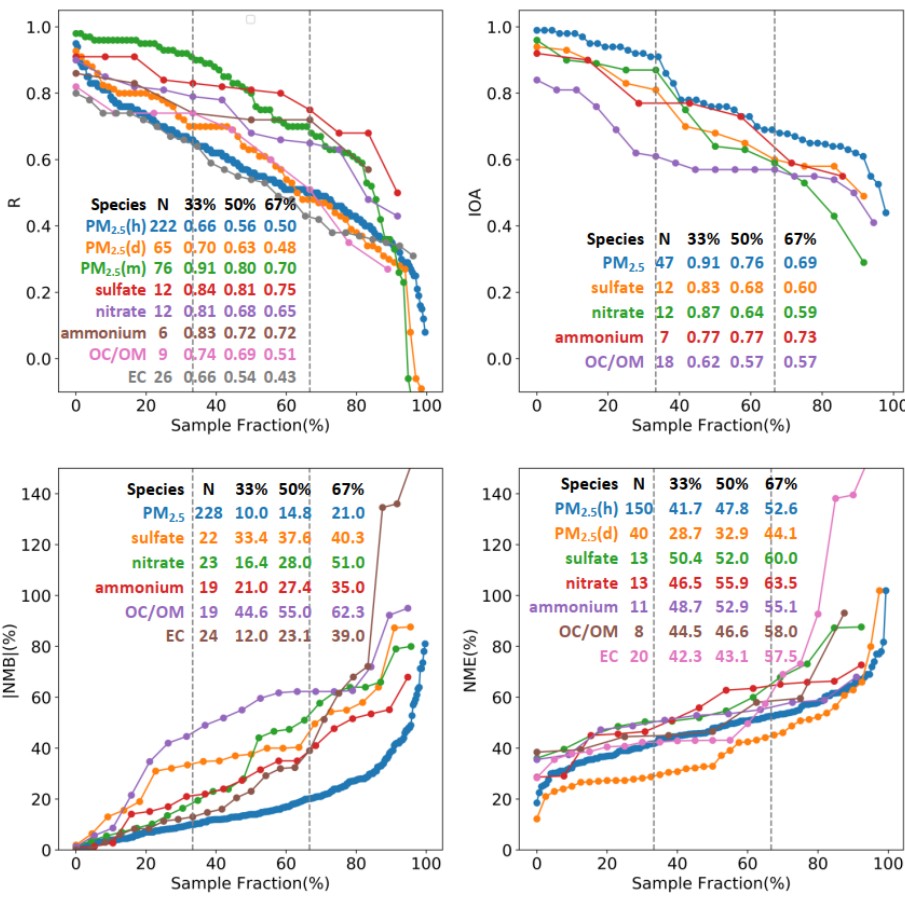

**Figure 9: Rank-ordered distributions of R, IOA, NMB and NME for total PM$_{2.5}$ and speciated components. The number of data points and the 33$^{rd}$, 50$^{th}$, and 67$^{th}$ percentile values are also listed.**





**Table 1 Definition of statistical metrics used in more than ten studies complied in this work**

| No. | Statistics (abbreviation) | Definition | Note |
|---|---|---|---|
| 1 | Correlation coefficient (R) | $\dfrac{\sum[(P_j - \bar{P}) \times (O_j - \bar{O})]}{\sqrt{\sum(P_j - \bar{P})^2 \times \sum(O_j - \bar{O})^2}}$ | Unitless, $-1 \leqslant R \leqslant 1$ |
| 2 | Index of agreement ($d$) | $1 - \dfrac{\sum(P_j - O_j)^2}{\sum(|P_j - \bar{O}| + |O_j - \bar{O}|)^2}$ | Unitless, $0 \leqslant d \leqslant 1$ |
| 3 | Normalize mean bias (NMB) | $\dfrac{\sum(P_j - O_j)}{\sum O_j} \times 100$ | $-100\% \leqslant NMB \leqslant +\infty$ |
| 4 | Normalize mean error (NME) | $\dfrac{\sum|P_j - O_j|}{\sum O_j} \times 100$ | $0\% \leqslant NME \leqslant +\infty$ |
| 5 | Fractional bias (FB) | $\dfrac{2}{N}\dfrac{\sum(P_j - O_j)}{(P_j + O_j)} \times 100$ | $-200\% \leqslant FB \leqslant +200\%$ |
| 6 | Fractional error (FE) | $\dfrac{2}{N}\dfrac{\sum|P_j - O_j|}{(P_j + O_j)} \times 100$ | $0\% \leqslant FE \leqslant +200\%$ |
| 7 | Root mean square error (RMSE) | $\sqrt{\dfrac{\sum(P_j - O_j)^2}{N}}$ | concentration unit |
| 8 | Mean bias (MB) | $\dfrac{\sum(P_j - O_j)}{N}$ | concentration unit |
| 9 | Mean error (ME) | $\dfrac{\sum|P_j - O_j|}{N}$ | concentration unit |

**Table 2: Recommended benchmarks for evaluating PGM applications in China for total PM$_{2.5}$ and speciated components [a, b]**

| Species | NMB | | NME | | R | | IOA | |
|---|---|---|---|---|---|---|---|---|
| | Goal | Criteria | Goal | Criteria | Goal | Criteria | Goal | Criteria |
| hourly PM$_{2.5}$ | $<\pm15\%$ | $<\pm25\%$ | $<45\%$ | $<55\%$ | $>0.60$ | $>0.45$ | $>0.90$ | $>0.65$ |
| daily PM$_{2.5}$ | $<\pm15\%$ | $<\pm25\%^*$ | $<30\%^*$ | $<45\%^*$ | $>0.65$ | $>0.45^*$ | $>0.90$ | $>0.65$ |
| monthly PM$_{2.5}$ | $<\pm15\%$ | $<\pm25\%$ | $<30\%$ | $<45\%$ | $>0.90$ | $>0.65$ | $>0.90$ | $>0.65$ |
| sulfate | $<\pm35\%$ | $<\pm45\%$ | $<55\%$ | $<65\%$ | $>0.80^*$ | $>0.70^*$ | $>0.80$ | $>0.60$ |
| nitrate | $<\pm20\%$ | $<\pm55\%^*$ | $<50\%^*$ | $<65\%^*$ | $>0.80$ | $>0.65$ | $>0.85$ | $>0.55$ |
| ammonium | $<\pm25\%$ | $<\pm40\%$ | $<50\%$ | $<60\%$ | $>0.80^*$ | $>0.70^*$ | $>0.75$ | $>0.70$ |
| OC/OM | $<\pm45\%$ | $<\pm65\%$ | $<45\%$ | $<60\%$ | $>0.70$ | $>0.50$ | $>0.60$ | $>0.50$ |
| EC | $<\pm15\%^*$ | $<\pm40\%$ | $<45\%^*$ | $<60\%^*$ | $>0.65$ | $>0.40$ | none | none |

[a] Values with an asterisk in Table 2 indicate that our benchmarks are stricter than corresponding values in Emery et al. (2017)
[b] Shaded values indicate that less than 10 data points were available to develop the benchmarks.

**Table 3: List of different formulas for index of agreement**

| Formula | Range | Reference |
|---|---|---|
| $d = 1 - \dfrac{\sum(P_j - O_j)^2}{\sum(|P_j - \bar{O}| + |O_j - \bar{O}|)^2}$ | [0,1] | Willmott (1981) |
| $d_1 = 1 - \dfrac{\sum|P_j - O_j|}{\sum(|P_j - \bar{O}| + |O_j - \bar{O}|)}$ | [0,1] | Willmott (1982) |
| $d_1' = 1 - \dfrac{\sum|P_j - O_j|}{2\sum|O_j - \bar{O}|)}$ | $(-\infty,1)$ | Willmott et al. (1985) |





$$d_r = \begin{cases} 1 - \dfrac{\sum|P_j - O_j|}{2\sum|O_j - \bar{O}|}, when \; \sum|P_j - O_j| \le 2\sum|O_j - \bar{O}| \\[3mm] \dfrac{2\sum|O_j - \bar{O}|}{2\sum|P_j - O_j|} - 1, when \; \sum|P_j - O_j| > 2\sum|O_j - \bar{O}| \end{cases}$$

[0,1]        Willmott et al. (2012)