# Peer review of "Recommendations on benchmarks for numerical air quality model applications in China: Part I – $PM_{2.5}$ and chemical species"

_Atmospheric Chemistry and Physics, 2020_

## Referee Comment (RC1) · Anonymous Referee #1 · 28 Jul 2020

Air pollution is a major environment problem and a hot scientific topic in China. Air quality model is a crucial kit to perform mechanism study, source apportionment study, strategy study and policy consultant. The usage of different air quality models increased exponentially over the past years. This work compiles studies during 2006-2019 using air quality models over China comprehensively, and analyses the accuracy of these studies over different regions with different models. Although the performance of some model results are compiled and evaluated in this work and the language presentation is good, however, I find this evaluation failed to follow the suggestion made by authors themselves and may be not based on a thoroughly review of previous modelling works. Furthermore, I find little improvement in this new reversion, it failed address my major

concerns in the quick review. I could not suggest for publishing the current version, unless the following concerns are well addressed.

1) A quick search on Web of Science tells me that there are about 74 papers published 2006-2019 using Geos-Chem to study air quality in China. This figure is much more than the other 3 models analysed in this study, CAMx, CMAQ, NAQPMS. Without include GEOS-Chem, I can not agree this samples used in this can represent the air quality modelling study in China and lead to a benchmark suggestion. Furthermore, I use the key word WRF-Chem, China and air quality, Web of Science gives me a result of 174 publications during 2006-2019. This figure is 3 times higher than the number of samples used in this study, which is only 56 samples. Authors need to fully justify the criteria them used for selecting samples.

2) The title does not reflect the present work. This work mainly focuses on PM, but the title highlights photochemical model. I feel more discussion about ozone pollution need to be included, given that ozone is the key secondary pollution of photochemistry and is becoming more and more important for air quality in China. Without including ozone, this study is far from any recommendation on benchmarks for photochemical models.

3) Authors need to include the evaluation of meteorology performs in this study, instead of "will be discussed as a future work". As suggested by authors themselves in the conclusion part: "It is always good practise to present model performance results of meteorological field. . . Performance results of meteorological model could also help explain potential causes of unsatisfactory PGM simulated results." Analyse the air quality performance in conjunction with meteorological performance will certainly improve the value of this work. Separating a nice and comprehensive work to individual pieces is not a good practise and also not good for a prestigious journal such as ACP.

4) As suggested by authors themselves in the conclusion part: "In addition to providing numerical values of statistical metrics for model performance evaluation, graphs/plots are strongly recommended to further support model validation. To give a few examples,

visualizing data via time series plots of modelled and observed data could help illustrate periods with better or poorer performances." I believe audiences are also expecting to see a time series plots of model performance over 2006-2019. Did we improve the ability of air quality simulation over past decades? If yes, what is the critical step we have improved; if no, where is key problem we should focus on in future? These are the key questions/suggestions we are keen to know from this comprehensive review study, and will add great value to this work and large help for the modelling community. However, this information is absent. I would like to suggest some further discuss in this direction, in addition to the summary of performance in previous works.

5) As suggested by authors themselves in the conclusion part: "Provide as much details as possible with respect to how observation and modelling results are used to obtain the statistical results." However, I feel very limited details are provided for some statistical analyses of this work. At lease, for me, it is difficult to understand or reproduce the Fig. 9. What does x-axis mean? "Sample fraction", fraction of what? Why the sum of fractions is larger than 100%, are they integrated values? Here is just an example, more details need to be provided in captions.
* * *

---

## Author Comment (AC1) · 1 Aug 2020

**Preliminary response to reviewer's comment on "Recommendations on benchmarks for photochemical grid model applications in China: Part I – PM2.5 and chemical species" by Ling Huang et al.**

Air pollution is a major environment problem and a hot scientific topic in China. Air quality model is a crucial kit to perform mechanism study, source apportionment study, strategy study and policy consultant. The usage of different air quality models increased exponentially over the past years. This work compiles studies during 2006-2019 using air quality models over China comprehensively, and analyses the accuracy of these studies over different regions with different models. Although the performance of some model results are compiled and evaluated in this work and the language presentation is good, however, I find this evaluation failed to follow the suggestion made by authors themselves and may be not based on a thoroughly review of previous modelling works. Furthermore, I find little improvement in this new reversion, it failed address my major concerns in the quick review. I could not suggest for publishing the current version, unless the following concerns are well addressed.

Response: In this short response, we reply to address the reviewer's comments in a quick and preliminary manner. A more detailed response with corresponding revisions will be provided separately.

1) A quick search on Web of Science tells me that there are about 74 papers published 2006-2019 using Geos-Chem to study air quality in China. This figure is much more than the other 3 models analysed in this study, CAMx, CMAQ, NAQPMS. Without include GEOS-Chem, I can not agree this samples used in this can represent the air quality modelling study in China and lead to a benchmark suggestion. Furthermore, I use the key word WRF-Chem, China and air quality, Web of Science gives me a result of 174 publications during 2006-2019. This figure is 3 times higher than the number of samples used in this study, which is only 56 samples. Authors need to fully justify the criteria them used for selecting samples.

Response: As suggested by the reviewer, we will include GEOS-Chem results in our revised manuscript. We will also provide a detailed description of our processes to select samples along with the revised manuscript. One thing we need to point out is that we started this work back in July 2019 so the results from Web of Science are expected to be a little different from what the reviewer found.

2) The title does not reflect the present work. This work mainly focuses on PM, but the title highlights photochemical model. I feel more discussion about ozone pollution need to be included, given that ozone is the key secondary pollution of photochemistry and is becoming more and more important for air quality in China. Without including ozone, this study is far from any recommendation on benchmarks for photochemical models.

Response: As specified in our current manuscript (Page 3, Line 17-20), we plan to prepare three companion papers: the first one (i.e. current one) focuses on $PM_{2.5}$ and speciated chemical components, considering that significant attention has been given

to PM$_{2.5}$ pollution in China for the past decade; the second one, which is currently under preparation, will be solely focusing on ozone, given that ozone pollution is becoming a more prominent problem over PM$_{2.5}$ in recent years; the last one will be focusing on other pollutants (e.g. PM$_{10}$, SO$_2$, NO$_2$). The purpose of this set of work is to give a comprehensive review of air quality model applications in China and the resulting model performance. We feel that it would be too much to include all information into one single manuscript. That's why we decided to present them separately.

3) Authors need to include the evaluation of meteorology performs in this study, instead of will be discussed as a future work . As suggested by authors themselves in the conclusion part: It is always good practise to present model performance results of meteorological field. . . Performance results of meteorological model could also help explain potential causes of unsatisfactory PGM simulated results. Analyse the air quality performance in conjunction with meteorological performance will certainly improve the value of this work. Separating a nice and comprehensive work to individual pieces is not a good practise and also not good for a prestigious journal such as ACP.

Response: We agree with the reviewer that meteorological performance is a critical part of a comprehensive and complete evaluation of air quality model application. However, for three reasons we decided to present meteorological evaluation results as a separate work. Firstly, the evaluation of meteorological modeling is a standalone scientific question by itself that requires a separate discussion for that. There are many more applications of meteorological simulations than providing inputs for air quality simulations. Secondly, as mentioned in our current manuscript (Page 5, Line 12), not all studies that performed evaluation of air quality simulations also evaluated meteorological simulations, given that good performance of air pollutants implicitly suggest accurate meteorological simulations. Lastly, including discussions on meteorological simulations would considerably increase the length of the current manuscript. Again, the current manuscript is aiming to focus on PM$_{2.5}$ and its chemical components. In summary, we acknowledge the importance of evaluating meteorological simulations and we feel it deserves a separate discussion.

4) As suggested by authors themselves in the conclusion part: In addition to providing numerical values of statistical metrics for model performance evaluation, graphs/plots are strongly recommended to further support model validation. To give a few examples visualizing data via time series plots of modelled and observed data could help illustrate periods with better or poorer performances. I believe audiences are also expecting to see a time series plots of model performance over 2006-2019. Did we improve the ability of air quality simulation over past decades? If yes, what is the critical step we have improved; if no, where is key problem we should focus on in future? These are the key questions/suggestions we are keen to know from this comprehensive review study, and will add great value to this work and large help for the modelling community. However, this information is absent. I would like to

suggest some further discuss in this direction, in addition to the summary of performance in previous works.

Response: We agree that graphical analysis is an important component of model performance evaluation. Graphical and statistical analyses are complementary. The reviewer provides relevant examples of how graphical analysis can be used to explain and illustrate important aspects of model performance. In response to the reviewer's suggestion, we can add a section that reviews best practices for using graphical analysis in model performance evaluation.

With respect to the time series plots of model performance over 2006-2019, we will look into this more and provide a detailed response in our upcoming revised manuscript.

5) As suggested by authors themselves in the conclusion part: Provide as much details as possible with respect to how observation and modelling results are used to obtain the statistical results. However, I feel very limited details are provided for some statistical analyses of this work. At lease, for me, it is difficult to understand or reproduce the Fig. 9. What does x-axis mean? Sample fraction, fraction of what? Why the sum of fractions is larger than 100%, are they integrated values? Here is just an example, more details need to be provided in captions.

Response: A short answer to questions regarding Figure 9. This is how we produce Figure 9. To give an example of IOA values reported for $PM_{2.5}$. There are in total 32 studies that reported IOA values for $PM_{2.5}$ and the total number of IOA reported is 47 (multiple IOA values could be reported in a single study). We sorted these 47 numbers from high to low and the corresponding sample fraction for individual number is calculated as the sorted rank divided by 47 (total number). Then we plot these 47 IOA numbers as y-axis and the corresponding sample fraction as x-axis (as shown in Figure 9). Based on this plot, we can directly tell one third (first dashed vertical line in Figure 9) of previously reported IOA values for $PM_{2.5}$ is greater than 0.91 and another third (second dashed vertical line) of previous reported IOA values is lower than 0.69. In this sense, the audiences could place their IOA results on Figure 9 and get a sense of where their results are located with respect to previous studies. We will provide more details in the upcoming revised manuscript.

---

## Referee Comment (RC2) · Anonymous Referee #3 · 6 Aug 2020

Severe air quality problem in China has recently attracted attention from the public as well as the scientific community. Photochemical grid models (PGMs) are frequently used to investigate the phenomenon and develop emission control strategies. The number of PGMs-based research articles for scientific and regulatory applications has surged. This study aimed to apply model performance evaluation (MPE) methodologies developed in U.S. to evaluate PGMs used in China for total PM2.5 and speciated PM components. A total of 128 recent peer reviewed articles based on one of four most popular PGMs were compiled and different model configurations as well as statistical metrics were evaluated. The benchmarks were developed and recommended for two tiers: "goals" and "criteria" for evaluating PGM applications in China. Although

the methodologies and metrics used in this study are not novel, or adopted from several studies conducted in U.S., the derived results/recommendations/conclusions are scientifically sound and its logic or context is reasonable. This study is expected to provide guidance for future PGM evaluations in China.

The manuscript is well written and the logic and context are well presented and easy to follow. Several comments are provided below in hope that these will assist the authors strengthen the manuscript.

Technical comments:

The methodologies and metrics adopted in this study are well established and published in several literature, and 128 relevant modeling studies conducted in China were compiled in this study for the model performance evaluation. Although the information provided in this study is useful for the modeling community, the analysis was relatively straightforward so I would consider this study a critical literature review, instead of a novel study. to strengthen the scope of this study, one would expect that the authors go beyond what was accomplished in the U.S. studies and consider additional analyses such as the following.

Although the manuscript describes the reasons why China specific modeling performance evaluation are needed, there are many commonalities across the air quality modeling community worldwide so comparison can be made among studies conducted in China or elsewhere. Emery et al (2017) indicates that "While we primarily address U.S. modeling and regulatory settings, these recommendations are relevant to any such applications of state-of-the-science photochemical models." The comparison of benchmarks from this study with Emery et al (2017) shows similarities. Thus, it seems that the benchmarks developed in China in this study confirm their worldwide applicability for other super-regional, regional, or local modeling domains. It would be valuable if the authors discuss the broader implication of these findings.

A total of 128 peer-reviewed articles were compiled for this study. Are there articles or

studies that were excluded from this study but could be potentially included by reapplying the metrics used in this study? Some of the studies may not report any MPE results but could be recalculated to get MPE results if needed. Please add some discussion on those studies, especially on those with speciated PM components since the number of these studies is very limited. If applicable, please include any additional studies so the dataset is larger or more meaningful. In addition to peer-reviewed articles, there may be non-peer-reviewed reports which deal with PGM applications (e.g., US EPA's PGM reports). I wonder if there are such reports published by Chinese central or provincial government or NGOs that can be included in this study.

On the other hand, are there cases (excluded in this study) that the authors can reapply the benchmarks recommended in this study to demonstrate the improved model robustness or validity? For example, there may be PGM studies that did not use the metrics adopted in this study but the evaluation may be improved after these metrics or benchmarks are applied.

Some benchmarks for speciated PM components are questionable due to the number of available studies, which may lead to biased or inconclusive results. Although caution is warranted, I wonder if the dataset can be enriched by including some studies elsewhere (e.g., U.S. studies) since benchmarks for speciated PM components were not studied in Emery et al (2017). I understand the focus of this study is in China, but it seems that benchmarks developed in China and U.S. are valid, comparable in both countries.

In-depth discussion on statistical metrics in "Impact of temporal and spatial resolution" (page 7, lines 35-37) is needed. This is counter-intuitive that the wider ranges are associated with the larger number of data points. What data is needed to improve the confidence on the benchmarks developed for speciated PM components?

Minor comments:

Page 5, line 29: Table 2 should be Table 1.

Page 6, line 40: please check the number. It seems one single study is in spring, not summer. There are 5 studies in summer.

Page 7, line 5-14: "Figure 6" is missing in this section and should appear somewhere.

Page 7, line 4 and 15: "Impact" is misspelled.

Page 7, line 33-34: it seems the R values correctly correspond to the coarsest resolution (80km) but off to the finest resolution (3km).

Page 8, line 5-13: it seems that the R values in the text correspond to different percentile. For instance, the 33rd percentile value should be 0.64 for hourly to 0.91 for monthly results while the 67th percentile should be 0.5 for hourly to 0.70 for monthly. Please check the remaining values in the text against Figure 9.

---

## Referee Comment (RC3) · Anonymous Referee #2 · 12 Aug 2020

The manuscript compiles 128 prior publications of chemical transport modeling studies on PM air pollution in China and summarizes model performances in commonly used statistics such as correlation, bias, quantile distribution, etc. I have three major concerns of the manuscript which makes it unsuitable for publication in ACP.

First, treating it as a research article I do not find the manuscript contains new knowledge in its current form. All the graphs and tables are simple summaries of the results from published papers. To justify their study, the authors make an affirmative statement in the introduction that benchmark metrics developed based on US and European studies may not be suitable for model evaluation in China (pg 7, line 5-7) but they do not

show scientific evidence or conduct their own analysis to support this claim. On the contrary, all the benchmark metrics the manuscript recommended have been proposed and used in the US or Europe and none of them is specific to China. The authors made an argument on the correlation coefficient being inconsistently used in prior studies (pg 4, line 10-15). I found this a trivial matter which can be easily reconciled by a careful reading of the reference of interest.

Second, treating it as a review article I do not find the manuscript conducts an objective and comprehensive review. It does not provide any justification for the selection criteria of publications included in the review. For example, what keywords did the authors use to search those 128 papers included in the manuscript? Why was the period of publication limited to be between 2006 and 2019? Why were only four models included?

Third, being a summary of prior modeling studies, the manuscript does not make any attempt to provide useful insights on why the published model performances on PM2.5 in China vary so much as shown in their figures. Is it due to different inventories, chemistry mechanisms, or meteorological fields used? Without this type of discussion, the manuscript would not provide much value to readers.
* * *

---

## Author Response (AR1)

Response to Reviewer's Comments

Anonymous Referee #1

Air pollution is a major environment problem and a hot scientific topic in China. Air quality model is a crucial kit to perform mechanism study, source apportionment study, strategy study and policy consultant. The usage of different air quality models increased exponentially over the past years. This work compiles studies during 2006-2019 using air quality models over China comprehensively, and analyses the accuracy of these studies over different regions with different models. Although the performance of some model results are compiled and evaluated in this work and the language presentation is good, however, I find this evaluation failed to follow the suggestion made by authors themselves and may be not based on a thoroughly review of previous modelling works. Furthermore, I find little improvement in this new reversion, it failed address my major concerns in the quick review. I could not suggest for publishing the current version, unless the following concerns are well addressed.

Response: We thank the reviewer for these valuable comments. We have made extensive efforts to improve the existing manuscript and aim to address the reviewer's concerns. Our major improvements include:

(1) re-write the Abstract/Introduction part to emphasize the importance and applications of results obtained from this work;

(2) adding GEOS-Chem results thus leading to a more complete picture of PGM applications in China;

(3) a detailed description of literature selection process is provided and the number of peer-reviewed publications increased from 128 to 307 (Section 2.1);

(4) more discussions were added to provide more in-depth insights into our findings, including trends of model performance results over the past decade (Section 3.2), application of Random Forest Model to rank feature importance of key model inputs

(Section 3.3 and Supplemental information).

Our responses to the reviewer's comment are given below in blue. Revised manuscript with revisions highlighted in yellow is attached after the response.

1) A quick search on Web of Science tells me that there are about 74 papers published 2006-2019 using Geos-Chem to study air quality in China. This figure is much more than the other 3 models analysed in this study, CAMx, CMAQ, NAQPMS. Without include GEOS-Chem, I can not agree this samples used in this can represent the air quality modelling study in China and lead to a benchmark suggestion. Furthermore, I use the key word WRF-Chem, China and air quality, Web of Science gives me a result of 174 publications during 2006-2019. This figure is 3 times higher than the number of samples used in this study, which is only 56 samples. Authors need to fully justify the criteria them used for selecting samples.

Response: As suggested by the reviewer, we included GEOS-Chem studies in our revised manuscript, which now leads to a total of five models included: CMAQ, CAMx, WRF-Chem, NAQPMS, and GEOS-Chem. The reason that we did not include GEOS-Chem at first is because GEOS-Chem is a global model with relatively coarse resolution (~200 km as default grid spacing), while the other four models are considered regional air quality models with relatively finer resolution (for example, 36 km and lower). Of the 20 GEOS-Chem studies that we compiled in this study, the finest grid resolution is $1/4° \times 5/16°$ (~30 km) and the coarsest resolution is $2° \times 2.5°$ (~200 km). As a result, GEOS-Chem cannot resolve details in the interactions between emissions and chemistry at city-scales and dispersion patterns will be rather different from the regional models. In the revised manuscript (Section 2.1), we also provide a detailed description of how we selected the samples from an initial 900+ Web of Science records down to a final set of 307 papers that were ultimately compiled in this study. This addresses the reviewer's concern regarding the criteria that we used for sample selection.

**2.1 Data compilation**

A total of five photochemical models – the Community Multiscale Air Quality (CMAQ, Foley et al., 2010), the Comprehensive Air Quality Model with Extensions (CAMx, Ramboll Environment and Health, 2018), the Goddard Earth Observing System (GEOS)-Chem (http://geos-chem.org), the Weather Research and Forecasting model coupled with Chemistry (WRF-Chem, Grell et al., 2005), and the Nested Air Quality Prediction Modelling System (NAQPMS, Z. Wang et al. 2006) – are included in this compilation. While the former four models are developed by institutes and/or companies outside China, the NAQPMS is developed by the Institute of Atmospheric Physics of Chinese Academy of Sciences and has mostly been utilized for applications in China.

GEOS-Chem is a global chemical transport model with coarser resolution (only 20% of complied GEOS-Chem studies has a grid resolution less than 50 km), as opposed to the other four regional models that are applied with finer spatial resolution at local scale (for example, less than 10 km). Our investigation started by searching for combinations of three key words on the Web of Science: model name, "air quality", and "China", and limited the timespan between 2006 and 2019. This initial search gives 446 (CMAQ), 84 (CAMx), 256 (WRF-Chem), 117 (NAQPMS), and 58 (GEOS-Chem) records (a total of 961). Duplicated records were excluded. We then excluded records that were listed as conference papers or not published in English-language journals (for example, Chinese and Korean-language journals). This resulted in 826 records published in 61 journals. We further reduced the number of journals considered by excluding those that had less than ten publications during 2006-2019, which results in 464 studies. Table S1 shows the list of journals that were included in this study, which is believed to cover the mainstream journals in atmospheric research, especially in applications of air quality models. The next filtering stage needed substantial manual effort. The 464 records were downloaded and manually checked to exclude (1) studies that were accidentally included in the search but did not apply any of the models in their study; (2) studies that were intended for other purposes (for example, evaluating meteorological simulations); (3) studies that were not focused on China (for example, the target region was Korea, Japan, etc.); (4) studies that did not provide any air quality model performance evaluation or the evaluation results were referred to previous studies; (5) studies that did conduct model performance evaluation but no numerical values were given (for example, only graphical plots were given). The final selection included a total of 307 papers (see a complete list in Table S2). We defined ten regions of China as shown in Figure 1, namely Beijing-Tianjin-Hebei (BTH) region, Yangtze River Delta (YRD) region, Pearl River Delta (PRD) region, Sichuan Basin (SCB), North China Plain (NCP), Central, Northwest, Northeast, Southeast, and Southwest (see Table S3 for provinces covered in this region).

2) The title does not reflect the present work. This work mainly focuses on PM, but the title highlights photochemical model. I feel more discussion about ozone pollution need to be included, given that ozone is the key secondary pollution of photochemistry and is becoming more and more important for air quality in China. Without including ozone, this study is far from any recommendation on benchmarks for photochemical models.

Response: As mentioned in our manuscript (Page 2, Line 34-37), we plan to develop three studies in series: the current study focuses on $PM_{2.5}$ and speciated chemical components (as stated in the title), considering that significant attention has been given to $PM_{2.5}$ pollution in China over the past decade. The second study, which is currently under preparation, will solely focus on ozone, given that ozone pollution is becoming a more prominent problem in recent years. The last study will focus on other pollutants (e.g. $PM_{10}$,

SO$_2$, NO$_2$). The purpose of this set of work is to give a comprehensive review of air quality model applications in China and their resulting performance against measurements. We feel that it would be too much to include all information in a single manuscript.

3) Authors need to include the evaluation of meteorology performs in this study, instead of will be discussed as a future work. As suggested by authors themselves in the conclusion part: It is always good practise to present model performance results of meteorological field. Performance results of meteorological model could also help explain potential causes of unsatisfactory PGM simulated results. Analyse the air quality performance in conjunction with meteorological performance will certainly improve the value of this work. Separating a nice and comprehensive work to individual pieces is not a good practise and also not good for a prestigious journal such as ACP.

Response: We agree with the reviewer that meteorological performance is a critical part of a comprehensive and complete evaluation of air quality model application. However, for three reasons we decided to present meteorological evaluation results as a separate work. First, the evaluation of meteorological modeling is a standalone scientific question by itself that requires a separate in-depth analysis and discussion. There are many more applications of meteorological simulations than providing inputs for air quality applications. Second, as mentioned in our current manuscript (Page 4, Line 40), not all studies that performed evaluation of their air quality simulations also evaluated or reported meteorological results, given that good air quality performance implicitly suggests acceptable meteorological simulations. Last, including discussions on meteorological evaluations would considerably increase the length of the current manuscript. Again, the current manuscript is aiming to focus on PM$_{2.5}$ and its chemical components as a way to guide future modeling applications with context about how well their modeling results compare to well-performing historical simulations specifically in China. In summary, we acknowledge the importance of evaluating meteorological simulations and we feel it deserves a separate discussion.

4) As suggested by authors themselves in the conclusion part: In addition to providing numerical values of statistical metrics for model performance evaluation, graphs/plots are strongly recommended to further support model validation. To give a few examples visualizing data via time series plots of modelled and observed data could help illustrate periods with better or poorer performances. I believe audiences are also expecting to see a time series plots of model performance over 2006-2019. Did we improve the ability of air quality simulation over past decades? If yes, what is the critical step we have improved; if no, where is key problem we should focus on in future? These are the key

questions/suggestions we are keen to know from this comprehensive review study, and will add great value to this work and large help for the modelling community. However, this information is absent. I would like to suggest some further discuss in this direction, in addition to the summary of performance in previous works.

Response: The reviewer brought up an excellent question: "Did we improve the ability of air quality simulation over past decades and if yes, what is the critical step that has been improved". The answer to this question is valuable for the modeling community, yet extremely difficult to answer, because model applications over the decades have been so diverse , with different and evolving models and physical/chemical treatments, different and evolving model inputs (e.g. emission inventory, boundary conditions, meteorological inputs), different model configurations (e.g. vertical layers, spatial resolutions), and different modeling periods and regions. These peer-reviewed studies were conducted independently and they are not designed as a set of controlled experiments to limit variability (i.e. varying one input/algorithm while holding others unchanged). Several studies were focused on improving model performance by developing better emission inventories. Some studies focused on model chemical mechanisms such as number of model species (SAPRC vs. carbon bond), incorporating new formation pathways (e.g. heterogeneous reactions, chlorine chemistry) or improving the modeling framework (two-product vs. VBS for SOA modeling). It would not be possible for us to analyze what factors influenced model performance in ~300 individual studies.

Nevertheless, we attempted to address the reviewer's comments as follows:

- To answer the reviewer's question, "Did we improve the ability of air quality simulation over past decades", we added time series plots of commonly used statistical metrics for $PM_{2.5}$ and speciated components (depending on data availability) reported in literature during 2006-2019 and associated discussions are added in the revised manuscript (see Section 3.2).

  In revised manuscript:

  ### *Trends over the past decade*

  In an attempt to assess whether model performance results have evolved over the past decades, we present time series of selected statistical metrics for total PM2.5 in Figure 9 (plots for inorganic species are shown in Figure S3). Results published prior to 2013 were aggregated into one group because there were a limited number of studies prior to 2013. For total $PM_{2.5}$, reported R values have remained relatively consistent over the past decade with the median fluctuating within 0.6~0.8. The ranges of reported RMSE and MB become narrower in recent years even though the number of studies has increased substantially. Reported IOA and RMSE values fluctuated upward and downward over the period. On the

other hand, there seems to be an improving trend in terms of FB, FE, and NME as the reported values for these three metrics shift towards zero. For instance, the median value of reported FE decreased from 56.9% prior to 2013 to around 33% in 2019. However, it is important not to over-interpret these results as the number of studies published each year could affect the results.

- In an effort to answer the reviewer's question, "what is the critical step we have improved", we did a preliminary analysis of 176 studies that reported model performance results for $PM_{2.5}$ based on the random forest method (see "Analysis of feature importance based on random forest method" in the revised supplemental information). Our results indicate that the top three factors involve emission inventory, grid resolution, and boundary conditions, while the choice of model and source of meteorology are least important. This is a very preliminary analysis and thus we have decided to include these discussions in the supporting materials.

In supplemental information:

**Analysis of feature importance based on Random Forest Method**

In this study, we applied the random forest method for pattern recognition to identify and rank model attributes (inputs, grid resolutions, etc.) that have important influences on $PM_{2.5}$ model performance. Random forest is a machine learning method suitable for classification and regression (Liu et al., 2012). It is a collection of a series of decision trees and each tree is generated from a bootstrap sample. Both continuous and categorical input variables are allowed. Like other machine learning methods, random forest is also a black box. It can provide the order of feature importance (FI) so that we can determine and rank which parameter choices most influence the simulation results.

We collected detailed model configurations for studies that reported results of correlation coefficient (R), index of agreement (IOA), mean bias (MB), normalized mean bias (NMB), mean error (ME), normalized mean error (NME), fractional bias (FB), and fraction error (FE) for $PM_{2.5}$ (a total of 176 studies). Model configurations include the meteorological data that are used to drive air quality simulations (e.g. from WRF, MM5, or GEOS), the emission inventory (e.g. public available dataset vs. locally developed), gas-phase chemistry (for example, carbon bond vs. SAPRC), aerosol chemistry (including inorganic aqueous chemistry, inorganic gas-particle partitioning, organic gas-particle partitioning and oxidation), boundary conditions (e.g. model default values vs. results generated from global model), grid resolution and the temporal resolution (Table S7). We did not include the study region and period for FI selection because we feel these two options are more restricted by the user's specific needs and focus (i.e., more subjective/uncontrollable and less objective/controllable). We ranked each statistical metric from good to poor performance. For example, values of R

and IOA that are close to 1 represent good performance and values close to 0 represent bad performances. For MB and NMB, we used absolute values so that deviations from zero represent the performance level. We then classified these results into three tiers with breaks at 33% and 67% of the ranked values so that each tier includes the top one third, the middle one third, and the bottom one third of the reported performance results. We then ran the random forest model using the 'sklearn' module in python to obtain the FI metric and the results are shown in Figure S4. The choice of emission inventory is shown to affect the model performances most, followed by grid resolution, aerosol and gas chemistry. Meteorological input and the choice of model itself is of least importance.

[Figure]

**Figure S4: Ranking of key model inputs in terms of feature importance**

- Finally, in addition to the model performance benchmarks, there also is a need for more studies that quantify contributions to model uncertainty, such as the recent study by Dunker et al. (2020), which quantifies the contributions of uncertainties associated with chemistry, boundary concentrations, deposition and emissions to uncertainty in simulated ozone results. These discussions were added in the revised manuscript (see Section 3.3).

In revised manuscript:

As mentioned earlier, PGM applications involve numerous driving inputs as well as diverse model configurations, which lead to an abundant database from which to assess their relative influences on model performance. A preliminary analysis based on the Random Forest Method (Liu et al., 2012), a machine learning method suitable for classification and regression, suggests that emission inventory, grid resolution and boundary conditions are the top three factors that affect model performances results (see details in Supplemental information). The similarities between the benchmarks derived in this study and Emery's study suggest that important model input data (e.g. emission inventories) have comparable accuracy for China and North America and model formulations (e.g. algorithms such as

chemistry, deposition, transport) seem to be equally applicable to China and North America. In additional to the need for model performance benchmarks, there also is a need for more studies that quantify contributions to model uncertainty, such as the recent study by Dunker et al. (2020), which quantifies contributions of chemistry, boundary concentrations, deposition and emissions to uncertainty in simulated ozone results.

5) As suggested by authors themselves in the conclusion part: Provide as much details as possible with respect to how observation and modelling results are used to obtain the statistical results. However, I feel very limited details are provided for some statistical analyses of this work. At lease, for me, it is difficult to understand or reproduce the Fig. 9. What does x-axis mean? Sample fraction, fraction of what? Why the sum of fractions is larger than 100%, are they integrated values? Here is just an example, more details need to be provided in captions.

Response: We have added more descriptions for Figure 10 in the revised manuscript. We also modified the caption to include more details. To give an example of IOA values reported for $PM_{2.5}$: there are in total 32 studies that reported IOA values for $PM_{2.5}$ and the total number of IOA values reported is 47 (multiple IOA values could be reported in a single study). We sorted these 47 numbers from high to low and calculated the corresponding sample fraction for each individual number as its sorted rank value divided by 47 (total number). Then we plot these 47 IOA numbers as y-axis and the corresponding sample fraction as x-axis (as shown in Figure 10). Based on this plot, we can directly tell that one third (first dashed vertical line in Figure 10) of previously reported IOA values for $PM_{2.5}$ is greater than 0.91 and another third (second dashed vertical line) of previous reported IOA values is lower than 0.69. In this sense, the reader can compare their IOA results to Figure 9 and get a sense of where their results are located with respect to previous studies. An example is added to the caption for clarification.

References:
Dunker, A.M., Wilson, G., Bates, J.T. and Yarwood, G., 2020. Chemical Sensitivity Analysis and Uncertainty Analysis of Ozone Production in the Comprehensive Air Quality Model with Extensions Applied to Eastern Texas. Environmental Science & Technology, 54(9), pp.5391-5399.
Liu, Y., Wang, Y., & Zhang, J. (2012, September). New machine learning algorithm: Random forest. In International Conference on Information Computing and Applications (pp. 246-252). Springer, Berlin, Heidelberg.

The manuscript compiles 128 prior publications of chemical transport modeling studies on PM air pollution in China and summarizes model performances in commonly used statistics such as correlation, bias, quantile distribution, etc. I have three major concerns of the manuscript which makes it unsuitable for publication in ACP.

Response: We thank the reviewer's comments. We have made substantial efforts to improve the existing manuscript and aim to address the reviewer's concerns. Our major revisions include:

(1) re-write the Abstract/Introduction part to emphasize the importance and applications of results obtained from this work;

(2) expand our compiled studies to include GEOS-Chem studies and provided a detailed description of how we compiled the studies;

(3) to provide discussions on how model performances evolve over the past 15 years; and

(4) to provide a preliminary analysis on the importance of several model key inputs on model performance results.

Our responses to the reviewer's comment are given below in blue. Revised manuscript with revisions highlighted in yellow is attached after the response.

First, treating it as a research article I do not find the manuscript contains new knowledge in its current form. All the graphs and tables are simple summaries of the results from published papers. To justify their study, the authors make an affirmative statement in the introduction that benchmark metrics developed based on US and European studies may not be suitable for model evaluation in China (pg 7, line 5-7) but they do not show scientific evidence or conduct their own analysis to support this claim. On the contrary, all the benchmark metrics the manuscript recommended have been proposed and used in

the US or Europe and none of them is specific to China. The authors made an argument on the correlation coefficient being inconsistently used in prior studies (pg 4, line 10-15). I found this a trivial matter which can be easily reconciled by a careful reading of the reference of interest.

Response: Applications of PGMs in China have increased significantly over the past decade because of their unique features and capabilities that cannot be achieved via field observations or chamber studies. PGM applications are especially important in air quality management practices because they are extensively used to identify source (both sectoral and regional) contributions to air quality problems as well as support formulation of control strategies. A critical step of all PGM applications is a robust and comprehensive model performance evaluation (MPE) to ensure the accuracy of the simulated results. However, MPE results from different studies differ dramatically. For example, the two most commonly used MPE statistical metrics – normalized mean bias (NMB) and correlation coefficient (R) could range -87%~110.6% and -0.59~0.98, respectively. No unified guidance, references or contextual information are provided for the ever-growing modeling community in China to interpret how well or bad their model performance results are.

Therefore, there are two objectives of this study. The first one is to provide a general overview of the status of PGM applications in China over the past 15 years; for example, the most frequently used models, the most frequently investigated regions, the most frequently used statistical metrics for model performances, etc. This is presented in Section 3.1. The second objective, which is more important, is to recommend performance benchmarks for commonly used statistical metrics (not new metrics!) based on studies in China. The underlying reasons for this need include: (1) no guidelines on systematic and standard model performance criteria or goals are available in China to provide context for good vs. poor results relative to the growing number of applications in China; and (2) the benchmarks that were used for quantitative model performance evaluation in some of the studies are based on results of PGM applications in the U.S., which are outdated (for example, the "Guidance on the Use of Models and Other Analyses for Demonstrating Attainment of Air Quality Goals for Ozone, $PM_{2.5}$ and Regional Haze" developed by the U.S. EPA was published in 2007 (EPA, 2007), which is almost 15 years ago; the benchmark of FB and FE for PM introduced by Boylan and Russell, which is used in quite some studies for model performance evaluation, was published in 2006 (Boylan and Russell, 2006)) and may not be appropriate for PGM applications in China, due to different model configurations (e.g. unique environments, quality of emission inventories, availability of source emission profiles, etc.). It is appropriate to come up with benchmarks that are solely based on PGM applications in China for a more direct apple-to-apple

comparison. The statistical metrics are not specific to China but the recommended benchmarks are. This part is presented in Section 3.3. This is the first time that a set of quantitative and objective MPE benchmarks that are suitable for PGM applications in China are recommended for the scientific and regulatory community. Results from the current study will support the ever-growing modelling community in China by allowing for objective assessments of how well their simulation results compare with historical studies and to better demonstrate the credibility and robustness of PGM applications prior to subsequent regulatory assessments. We have re-written the abstract and introduction part to emphasize the importance of this work.

[revised manuscript text omitted]

We mentioned that "correlation coefficient being inconsistently used in prior studies" is simply an example of the situation where a same statistical metrics could be calculated in different ways and cautions need to be taken when interpreting the performance results.

Second, treating it as a review article I do not find the manuscript conducts an objective and comprehensive review. It does not provide any justification for the selection criteria of publications included in the review. For example, what keywords did the authors use to search those 128 papers included in the manuscript? Why was the period of publication limited to be between 2006 and 2019? Why were only four models included?

Response: We thank the reviewer for this question. As suggested by the other reviewers, we added GEOS-Chem studies, thus leading to a total of five models being investigated. We are confident that these five models represent the mainstream air quality models used in China. In the revised manuscript, we added a full description (Section 2.1) of how we conducted the selection process from the initial search on Web of Science, to screening an initial of 900+ studies down to 307 studies. The period of 2006 to 2019 was selected because few studies used air quality models prior to 2006.

In revised manuscript:

A total of five photochemical models – the Community Multiscale Air Quality (CMAQ, Foley et al., 2010), the Comprehensive Air Quality Model with Extensions (CAMx, Ramboll Environment and Health, 2018), the Goddard Earth Observing System (GEOS)-Chem (http://geos-chem.org), the Weather Research and Forecasting model coupled with Chemistry (WRF-Chem, Grell et al., 2005), and the Nested Air Quality Prediction Modelling System (NAQPMS, Z. Wang et al. 2006) – are included in this compilation. While the former four models are developed by institutes and/or companies outside China, the NAQPMS is developed by the Institute of Atmospheric Physics of Chinese Academy of Sciences and has mostly been utilized for applications in China. GEOS-Chem is a global chemical transport model with coarser resolution (only 20% of complied GEOS-Chem studies has a grid resolution less than 50 km), as opposed to the other four regional models that are applied with finer spatial resolution at local scale (for example, less than 10 km). Our investigation started by searching for combinations of three key words on the Web of Science: model name, "air quality", and "China", and limited the timespan between 2006 and 2019. This initial search gives 446 (CMAQ), 84 (CAMx), 256 (WRF-Chem), 117 (NAQPMS), and 58 (GEOS-Chem) records (a total of 961). Duplicated records were excluded. We then excluded records that were listed as conference papers or not published in English-language journals (for example, Chinese and Korean-language journals). This resulted in 826 records published in 61 journals. We further reduced the number of journals considered by excluding those that had less than ten publications during 2006-2019, which results in 464 studies. Table S1 shows the list of journals that were included in this study, which is believed to cover the mainstream journals in atmospheric research, especially in applications of air quality models. The next filtering stage needed substantial manual effort. The 464 records were downloaded and manually checked to

exclude (1) studies that were accidentally included in the search but did not apply any of the models in their study; (2) studies that were intended for other purposes (for example, evaluating meteorological simulations); (3) studies that were not focused on China (for example, the target region was Korea, Japan, etc.); (4) studies that did not provide any air quality model performance evaluation or the evaluation results were referred to previous studies; (5) studies that did conduct model performance evaluation but no numerical values were given (for example, only graphical plots were given). The final selection included a total of 307 papers (see a complete list in Table S2). We defined ten regions of China as shown in Figure 1, namely Beijing-Tianjin-Hebei (BTH) region, Yangtze River Delta (YRD) region, Pearl River Delta (PRD) region, Sichuan Basin (SCB), North China Plain (NCP), Central, Northwest, Northeast, Southeast, and Southwest (see Table S3 for provinces covered in this region).

Third, being a summary of prior modeling studies, the manuscript does not make any attempt to provide useful insights on why the published model performances on PM2.5 in China vary so much as shown in their figures. Is it due to different inventories, chemistry mechanisms, or meteorological fields used? Without this type of discussion, the manuscript would not provide much value to readers.

Response: The underlying reason for diverse model performance of $PM_{2.5}$ is an important question, but yet difficult to answer. As a group, these ~300 studies utilized diverse input data (e.g. emission inventory, boundary conditions, meteorological inputs) and model formulations (e.g. gas- and aerosol chemistry), and spatial resolutions. On top of this, these studies focused on different modeling periods and regions. Therefore, they were not designed as a set of controlled experiments (i.e., varying one input/algorithm while holding the others constant) with an objective of investigating sources of model uncertainty. In addition to that, because obtaining "good model performance" is an essential requirement for publishing, it is likely that researchers have already tested alternative model configurations (e.g., with different meteorology, emissions, chemistry) before choosing and publishing the best-performing base-case results (Reynolds et al., 1996). Consequently, errors in inputs and algorithms present in this group of simulations are likely to be correlated (e.g., tendency for higher emissions correlated with more dispersive meteorological simulations), which will hinder reliable diagnosis of factors contributing to model error. Nevertheless, we tried our best to address these questions as follows:

- First, we added time series plots of commonly used statistical metrics for $PM_{2.5}$ and speciated components (depending on data availability) reported in the literature reviewed herein to illustrate an overall trend of model performance during the past decade (see Section 3.2)

  In revised manuscript:

*Trends over the past decade*

In an attempt to assess whether model performance results have evolved over the past decades, we present time series of selected statistical metrics for total PM2.5 in Figure 9 (plots for inorganic species are shown in Figure S3). Results published prior to 2013 were aggregated into one group because there were a limited number of studies prior to 2013. For total $PM_{2.5}$, reported R values have remained relatively consistent over the past decade with the median fluctuating within 0.6~0.8. The ranges of reported RMSE and MB become narrower in recent years even though the number of studies has increased substantially. Reported IOA and RMSE values fluctuated upward and downward over the period. On the other hand, there seems to be an improving trend in terms of FB, FE, and NME as the reported values for these three metrics shift towards zero. For instance, the median value of reported FE decreased from 56.9% prior to 2013 to around 33% in 2019. However, it is important not to over-interpret these results as the number of studies published each year could affect the results.

- Second, we did a preliminary analysis of 176 studies that reported model performance results for $PM_{2.5}$ using the Random Forest Method to rank the influence of key model attributes such as inputs, chemical and dispersion formulations, and grid resolution (see "Analysis of feature importance based on Random Forest Method" in revised supplemental information). Our results indicate that emission inventory, grid resolution, and boundary conditions are the top three factors influencing good vs. poor model results, while the choice of model and meteorology are least important. This is a very preliminary analysis and thus we decided to include these results in the supporting materials.

  In supplemental information:

**Analysis of feature importance based on Random Forest Method**

In this study, we applied the random forest method for pattern recognition to identify and rank model attributes (inputs, grid resolutions, etc.) that have important influences on $PM_{2.5}$ model performance. Random forest is a machine learning method suitable for classification and regression (Liu et al., 2012). It is a collection of a series of decision trees and each tree is generated from a bootstrap sample. Both continuous and categorical input variables are allowed. Like other machine learning methods, random forest is also a black box. It can provide the order of feature importance (FI) so that we can determine and rank which parameter choices most influence the simulation results.

We collected detailed model configurations for studies that reported results of correlation coefficient (R), index of agreement (IOA), mean bias (MB), normalized mean bias (NMB), mean error (ME), normalized mean error (NME), fractional bias (FB), and fraction error (FE)

for PM$_{2.5}$ (a total of 176 studies). Model configurations include the meteorological data that are used to drive air quality simulations (e.g. from WRF, MM5, or GEOS), the emission inventory (e.g. public available dataset vs. locally developed), gas-phase chemistry (for example, carbon bond vs. SAPRC), aerosol chemistry (including inorganic aqueous chemistry, inorganic gas-particle partitioning, organic gas-particle partitioning and oxidation), boundary conditions (e.g. model default values vs. results generated from global model), grid resolution and the temporal resolution (Table S7). We did not include the study region and period for FI selection because we feel these two options are more restricted by the user's specific needs and focus (i.e., more subjective/uncontrollable and less objective/controllable). We ranked each statistical metric from good to poor performance. For example, values of R and IOA that are close to 1 represent good performance and values close to 0 represent bad performances. For MB and NMB, we used absolute values so that deviations from zero represent the performance level. We then classified these results into three tiers with breaks at 33% and 67% of the ranked values so that each tier includes the top one third, the middle one third, and the bottom one third of the reported performance results. We then ran the random forest model using the 'sklearn' module in python to obtain the FI metric and the results are shown in Figure S4. The choice of emission inventory is shown to affect the model performances most, followed by grid resolution, aerosol and gas chemistry. Meteorological input and the choice of model itself is of least importance.

[Figure]

**Figure S4: Ranking of key model inputs in terms of feature importance**

- Last but not the least, separate from the need for model performance benchmarks, there is also a need for more studies that quantify contributions to model uncertainty, such as the recent study by Dunker et al. (2020), which quantifies contributions of chemistry, boundary concentrations, deposition and emissions to uncertainty in ozone model results. These discussions were added in Section 3.3.
  In revised manuscript:

As mentioned earlier, PGM applications involve numerous driving inputs as well as diverse model configurations, which lead to an abundant database from which to assess their relative influences on model performance. A preliminary analysis based on the Random Forest Method (Liu et al., 2012), a machine learning method suitable for classification and regression, suggests that emission inventory, grid resolution and boundary conditions are the top three factors that affect model performances results (see details in Supplemental information). The similarities between the benchmarks derived in this study and Emery's study suggest that important model input data (e.g. emission inventories) have comparable accuracy for China and North America and model formulations (e.g. algorithms such as chemistry, deposition, transport) seem to be equally applicable to China and North America. In additional to the need for model performance benchmarks, there also is a need for more studies that quantify contributions to model uncertainty, such as the recent study by Dunker et al. (2020), which quantifies contributions of chemistry, boundary concentrations, deposition and emissions to uncertainty in simulated ozone results.

Our responses to the reviewer's comment are given below in blue. Revised manuscript with revisions highlighted in yellow is attached after the response.

Technical comments:

The methodologies and metrics adopted in this study are well established and published in several literature and 128 relevant modeling studies conducted in China were compiled in this study for the model performance evaluation. Although the information provided in this study is useful for the modeling community, the analysis was relatively straightforward so I would consider this study a critical literature review, instead of a novel study. To strengthen the scope of this study, one would expect that the authors go beyond what was accomplished in the U.S. studies and consider additional analyses such as the following.

Although the manuscript describes the reasons why China specific modeling performance evaluation are needed, there are many commonalities across the air quality modeling community worldwide so comparison can be made among studies conducted in China or elsewhere. Emery et al (2017) indicates that "While we primarily address U.S. modeling and regulatory settings, these recommendations are relevant to any such applications of state-of-the-science photochemical models." The comparison of benchmarks from this study with Emery et al (2017) shows similarities. Thus, it seems that the benchmarks developed in China in this study confirm their worldwide applicability for other super-regional, regional, or local modeling domains. It would be valuable if the authors discuss the broader implication of these findings.

Response: The benchmarks results are updated based on the updated compilation of studies. Currently, no guidelines on systematic and standard model performance criteria or goals are available in China to provide context for good vs. poor results relative to the growing number of applications in China. Benchmarks that were used for quantitative model performance evaluation in some of the studies are based on results of PGM applications in the U.S., some of which are outdated (for example, the "Guidance on the Use of Models and Other Analyses for Demonstrating Attainment of Air Quality Goals for Ozone, $PM_{2.5}$ and Regional Haze" developed by the U.S. EPA was published in 2007 (EPA, 2007), which is almost 15 years ago; the benchmark of FB and FE for PM introduced by Boylan and Russell, which is used in quite some studies for model

performance evaluation, was published in 2006 (Boylan and Russell, 2006)) and may not be appropriate for PGM applications in China, due to different model configurations (e.g. unique environments, quality of emission inventories, availability of source emission profiles, etc.). It is appropriate to come up with benchmarks that are solely based on PGM applications in China for a more direct apple-to-apple comparison. More discussions with respect to the broader implications have been added to the revised manuscript in Section 3.3:

In revised manuscript:

As mentioned earlier, PGM applications involve numerous driving inputs as well as diverse model configurations, which lead to an abundant database from which to assess their relative influences on model performance. A preliminary analysis based on the Random Forest Method (Liu et al., 2012), a machine learning method suitable for classification and regression, suggests that emission inventory, grid resolution and boundary conditions are the top three factors that affect model performances results (see details in Supplemental information). The similarities between the benchmarks derived in this study and Emery's study suggest that important model input data (e.g. emission inventories) have comparable accuracy for China and North America and model formulations (e.g. algorithms such as chemistry, deposition, transport) seem to be equally applicable to China and North America. In additional to the need for model performance benchmarks, there also is a need for more studies that quantify contributions to model uncertainty, such as the recent study by Dunker et al. (2020), which quantifies contributions of chemistry, boundary concentrations, deposition and emissions to uncertainty in simulated ozone results.

A total of 128 peer-reviewed articles were compiled for this study. Are there articles or studies that were excluded from this study but could be potentially included by reapplying the metrics used in this study? Some of the studies may not report any MPE results but could be recalculated to get MPE results if needed. Please add some discussion on those studies, especially on those with speciated PM components since the number of these studies is very limited. If applicable, please include any additional studies so the dataset is larger or more meaningful. In addition to peer-reviewed articles, there may be non-peer-reviewed reports which deal with PGM applications (e.g., US EPA's PGM reports). I wonder if there are such reports published by Chinese central or provincial government or NGOs that can be included in this study.

Response: In our revised manuscript, we expanded our data compilation to include (1) studies that applied GEOS-Chem and (2) studies that were published in late 2019. We added a detailed description of how we selected the samples from initially 900+ Web of Science records down to a final of 307 papers used in this study. Studies that are excluded in our compilation are studies that either do not report model performance

results or results are presented in graphical format (we did not want to "estimate" values from figures). It was just not feasible to recalculate MPE metrics without having the source data (i.e. simulated results and observations used). The number of studies that reported MPE results for speciated PM components has been increased to 169 studies.

As far as we understand, there are no official reports published by Chinese central or provincial governments that address PGM applications, especially with respect to model performance results. The "Manual for Compilation of Urban Air Quality Standards Planning[1]" is issued by the Clean Air Innovation Center, the Ministry of Environmental Protection, the Chinese Academy of Environmental Sciences, Tsinghua University and other institutions with the purpose of guiding cities in the preparation of air quality compliance plans and to establish a systematic air quality compliance management model accordingly. The manual includes the specific methods and steps for the preparation of the plan to achieve the standards, and air quality models (including CMAQ, CAMx, WRF-Chem, NAQPMS) were only briefly recommended as related tools to help the city complete the preparation of the air quality plan.

On the other hand, are there cases (excluded in this study) that the authors can reapply the benchmarks recommended in this study to demonstrate the improved model robustness or validity? For example, there may be PGM studies that did not use the metrics adopted in this study but the evaluation may be improved after these metrics or benchmarks are applied.

Response: It is difficult to answer this question, given that any response here or in the manuscript would be strictly hypothetical given that we don't know what those cases entail that are excluded from this study. The point of this work and the recommended benchmarks is to provide context so that future modelers understand more directly and objectively where their simulation results stand in the universe of Chinese PGM applications. There may be some historical studies that can benefit from this new information, especially those that comprise the outer 67% of results reported here as they would now know that their simulations need to be reworked to improve performance. But it is difficult to say that the benchmarks would provide evidence for directions in which to pursue improvements as there are just too many variables involved.

Some benchmarks for speciated PM components are questionable due to the number of available studies, which may lead to biased or inconclusive results. Although caution is warranted, I wonder if the dataset can be enriched by including some studies elsewhere (e.g., U.S. studies) since benchmarks for speciated PM components were not studied in Emery et al (2017). I understand the focus of this study is in China, but it seems that
* * *
[1] https://www.efchina.org/Reports-zh/report-cemp-20170928-zh

benchmarks developed in China and U.S. are valid, comparable in both countries.

Response: We have expanded the number of studies to 307 and thus the resulting dataset for speciated PM components are enriched substantially. To be consistent with total $PM_{2.5}$ as well as follow-up work on ozone and other criteria pollutants, we decide to only use results from PGM applications in China.

In-depth discussion on statistical metrics in "Impact of temporal and spatial resolution" (page 7, lines 35-37) is needed. This is counter-intuitive that the wider ranges are associated with the larger number of data points. What data is needed to improve the confidence on the benchmarks developed for speciated PM components?

Response: We included GEOS-Chem studies in the revised manuscript and the spatial resolution range from as fine as 1km to as coarse as over 100 km. Instead of presenting results by specific resolution, we grouped results into five categories of spatial resolution range: (0, 5 km], (5 km, 10 km], (10 km, 25 km], (25, 50 km], and (50 km, 100 km]. It is not necessary that winder ranges are associated with large number of data points. For example, reported values for (10, 25 km] is associated widest range but not necessarily the large number of data points. As mentioned in the manuscript, there are a lot of parameters that could influence the model performance results. Based on our analysis of feature importance using the random forest method (see "Analysis of feature importance based on Random Forest Method" in Supplemental information), grid resolution turns out to be the second most important factor for model performances. These discussions have been added to the revised manuscript.

In revised manuscript:

**Impact of temporal and spatial resolution**

[revised manuscript text omitted]

generated from a bootstrap sample. Both continuous and categorical input variables are allowed. Like other machine learning methods, random forest is also a black box. It can provide the order of feature importance (FI) so that we can determine and rank which parameter choices most influence the simulation results.

We collected detailed model configurations for studies that reported results of correlation coefficient (R), index of agreement (IOA), mean bias (MB), normalized mean bias (NMB), mean error (ME), normalized mean error (NME), fractional bias (FB), and fraction error (FE) for $PM_{2.5}$ (a total of 176 studies). Model configurations include the meteorological data that are used to drive air quality simulations (e.g. from WRF, MM5, or GEOS), the emission inventory (e.g. public available dataset vs. locally developed), gas-phase chemistry (for example, carbon bond vs. SAPRC), aerosol chemistry (including inorganic aqueous chemistry, inorganic gas-particle partitioning, organic gas-particle partitioning and oxidation), boundary conditions (e.g. model default values vs. results generated from global model), grid resolution and the temporal resolution (Table S7). We did not include the study region and period for FI selection because we feel these two options are more restricted by the user's specific needs and focus (i.e., more subjective/uncontrollable and less objective/controllable). We ranked each statistical metric from good to poor performance. For example, values of R and IOA that are close to 1 represent good performance and values close to 0 represent bad performances. For MB and NMB, we used absolute values so that deviations from zero represent the performance level. We then classified these results into three tiers with breaks at 33% and 67% of the ranked values so that each tier includes the top one third, the middle one third, and the bottom one third of the reported performance results. We then ran the random forest model using the 'sklearn' module in python to obtain the FI metric and the results are shown in Figure S4. The choice of emission inventory is shown to affect the model performances most, followed by grid resolution, aerosol and gas chemistry. Meteorological input and the choice of model itself is of least importance.

[Figure]

**Figure S4: Ranking of key model inputs in terms of feature importance**

The confidence on the benchmarks developed for speciated PM components are mostly limited by the number of available data points. Therefore, more results from studies that perform model validation against observed speciated PM would be needed to improve the confidence on developed benchmarks.

Minor comments:

Page 5, line 29: Table 2 should be Table 1.

Response: Corrected in revised manuscript.

Page 6, line 40: please check the number. It seems one single study is in spring, not summer. There are 5 studies in summer.

Response: Since the underlying dataset has been updated with more studies, this comment is not relevant.

Page 7, line 5-14: "Figure 6" is missing in this section and should appear somewhere.

Response: Added in the manuscript (Page 6, Line 39).

Page 7, line 4 and 15: "Impact" is misspelled.

Response: Corrected in revised manuscript.

Page 7, line 33-34: it seems the R values correctly correspond to the coarsest resolution (80km) but off to the finest resolution (3km).

Response: We have revised Figure 8 with updated results and grouped different resolutions into four categories. Thus this comment is not relevant.

Page 8, line 5-13: it seems that the R values in the text correspond to different percentile. For instance, the 33rd percentile value should be 0.64 for hourly to 0.91 for monthly results while the 67th percentile should be 0.5 for hourly to 0.70 for monthly. Please check the remaining values in the text against Figure 9.

Response: The reviewer is correct about this. We mistakenly reversed the values of 33$^{rd}$ and 67$^{th}$ percentile. We have corrected this in the revised manuscript with updated results.

References:

Boylan, J. W., & Russell, A. G. (2006). PM and light extinction model performance metrics, goals, and criteria for three-dimensional air quality models. Atmospheric Environment, 40(26), 4946-4959.

EPA, U. (2007). Guidance on the use of models and other analyses for demonstrating attainment of air quality goals for ozone, PM2.5, and regional haze. US Environmental Protection Agency, Office of Air Quality Planning and Standards.

*Revised manuscript*

[revised manuscript text omitted]

---

## Author Response (AR2)

**Point-by-point Response to Reviewer's Comments**

**Report #1 by Anonymous Referee #3**

The revised manuscript has improved its scientific integrity and responded to questions/comments provided during the interactive discussion. Here are a few additional comments:

We appreciate the reviewer for taking time to carefully review the manuscript and give detailed and constructive comments, which has greatly helped to improve this paper. Below is our point-by-point response to each comment.

• Page 1 (Abstract): Abstract should include major results or conclusions derived from this study.
**Response**: We have revised the abstract by including major results of this study, which has been highlighted in the revised manuscript and also shown below.

**Revised abstract:**
Numerical air quality models (AQMs) are being applied more frequently over the past decade to address diverse scientific and regulatory issues associated with deteriorated air quality in China. Thorough evaluation of a model's ability to replicate monitored conditions (i.e. a model performance evaluation or MPE) helps to illuminate the robustness and reliability of the baseline modelling results and subsequent analyses. However, with numerous input data requirements, diverse model configurations, and the scientific evolution of the models themselves, no two AQM applications are the same and their performance results should be expected to differ. MPE procedures have been developed for Europe and North America but there is currently no uniform set of MPE procedures and associated benchmarks for China. Here we present an extensive review of model performance for fine particulate matter ($PM_{2.5}$) AQM applications to China and, from this context, propose a set of statistical benchmarks that can be used to objectively evaluate model performance for $PM_{2.5}$ AQM applications in China. We compiled MPE results from 307 peer-reviewed articles published between 2006 and 2019, which applied five of the most frequently used AQMs in China. We analyse influences on the range of reported statistics from different model configurations, including modelling regions and seasons, spatial resolution of modelling grids, temporal resolution of the MPE, etc. Analysis using a Random Forest method shows that the choices of emission inventory, grid resolution, and aerosol and gas-phase chemistry are the top three factors affecting model performance for $PM_{2.5}$. We propose benchmarks for six frequently used evaluation metrics for AQM applications in China, including two tiers – "goals" and "criteria" – where "goals" represent the best model performance that a model is currently expected to achieve and "criteria" represent the model performance that the majority of studies can meet. Our results formed a benchmark framework for the modelling performance of $PM_{2.5}$ and its chemical species in China. For instance, in order to meet the goal and criteria, the normalized mean bias (NMB) for total $PM_{2.5}$ should be within 10% and 20% while the normalized mean error (NME) should be within 35% and 45%, respectively. The goal and criteria values of correlation coefficients for evaluating hourly and daily $PM_{2.5}$ are 0.70 and 0.60, respectively; corresponding values are higher when the index of agreement (IOA) is used (0.80 for goal and 0.70 for criteria). Results from this study will support the ever-growing modelling community in China by

providing a more objective assessment and context for how well their results compare with previous studies, and to better demonstrate the credibility and robustness of their AQM applications prior to subsequent regulatory assessments.

• Page 3 (2.1 Data compilation): The number of studies considered in this revision increased significantly from 128 to 307, which is astounding. Since this is the first of a series of PGM evaluation studies, it is critical that a set of selection criteria is well defined and strictly applied. The five selection criteria used in this study are scientifically sounds but excluding non-English journals or journals with <10 publications is not. In particular, the latter seems to have over 300 publications available and relevant to the PGM evaluation but excluded in this study.

**Response**: We agree with the reviewer that a clear description of the set of selection criteria is important before demonstrating the results. A detailed description of the selection criteria was provided in the manuscript. With these criteria strictly applied, we finally included 307 articles in this study, which is much larger than previous similar studies for U.S. (e.g. only 69 studies in Simon et al. (2012) and 76 studies in Emery et al. (2017)). We believe that this compilation could give a general picture of the model performances.

We excluded the non-English journals because: (1) compared to English journals, they have narrower audiences; (2) the majority of the evaluation results covered by the non-English journals are also covered by the English journals written by the same group of researchers; and (3) we believe that most of the evaluation results reported by the Chinese journals are comparable with those published in English.

We excluded journals with less than 10 publications for the following reasons: (1) The 307 studies (out of 464 studies found by our Web of Science search) that we included are published in main stream air quality-related journals (especially in the field of air quality modeling). In contrast, many of the excluded studies were not in air quality-related journals and they appeared in the Web of Science search simply because of a key word. (2) 307 studies is a large body of data from which to draw conclusions suggesting that adding more studies is statistically unlikely to change our findings. In summary, we believe that the 307 included studies are representative of current results for air quality model applications and including more studies would be unlikely to change our major conclusions. We revised the manuscript and inserted the explanations.

**Revised manuscript** (Page 3, Line 16-20):
Our investigation started by searching for combinations of three key words on the Web of Science: model name, "air quality", and "China", and limited the timespan between 2006 and 2019. This initial search gave 446 (CMAQ), 84 (CAMx), 256 (WRF-Chem), 117 (NAQPMS), and 58 (GEOS-Chem) records (a total of 961). Duplicated records were excluded. We then excluded records that were listed as conference papers or not published in English-language journals (for example, Chinese and Korean-language journals) due to narrower audiences. This resulted in 826 records published in 61 journals. We further reduced the number of journals considered by excluding those that had less than ten publications during 2006-2019, since most of the excluded journals are not air quality-related journals, which results in 464 studies. Table S1 shows the list of journals that were included in this study, which is believed to cover the mainstream journals in atmospheric research, especially in applications of air quality models.

• Page 8 (3.3. Recommended metrics and benchmarks): Figure 10 shows 6 graphs in order of NMB, NME, FB, FE, R, and IOA while the text discusses the Figure in order of R, IOA, NMB, NME, FB and FE. Can the order be consistent? It seems that some numbers have not been updated from the previous version so please check. Most of the values in the text reflect what is shown in Figure 10 but some numbers rounded.

**Response**: Thanks to the reviewer for pointing out this inconsistent issue. The order of Figure 10 and Table 2 are revised to match text discussions. We double checked the values. We rounded up some numbers to nearest 0.5 or 5% to consistently recommend a set of round values.

**Revised figure and table**:

[Figure]

**Figure 10**:**Rank-ordered distributions of R, IOA, NMB, NME, FB, and FE for total PM$_{2.5}$ and speciated components. The number of data points and the 33$^{rd}$, 50$^{th}$, and 67$^{th}$ percentile values are also listed. For instance, one third of reported R value for predicted hourly PM$_{2.5}$ concentration is higher than 0.76; half is higher than 0.69; and two thirds higher than 0.60.**

**Table 2: Recommended benchmarks for evaluating AQM applications in China for total PM$_{2.5}$ and speciated components [a, b]**

| Metrics | Benchmark level | PM$_{2.5}$ | sulfate | nitrate | ammonium | OC/OM | EC |
|---|---|---|---|---|---|---|---|
| **R** | **Goal** | >0.70 (hourly/daily) >0.90 (monthly) | >0.75[*] | >0.70 | >0.75[*] | >0.65 | >0.65 |
| | **Criteria** | >0.60[*] (hourly/daily) | >0.65[*] | >0.60 | >0.65[*] | >0.55 | >0.45 |

| | | >0.70 (monthly) | | | | | |
|---|---|---|---|---|---|---|---|
| **IOA** | **Goal** | >0.80 | >0.80 | >0.85 | >0.75 | >0.75 | None |
| | **Criteria** | >0.70 | >0.60 | >0.50 | >0.60 | >0.55 | None |
| **NMB** | **Goal** | <±10% | <±20% | <±20% | <±15% | <±35% | <±20% |
| | **Criteria** | <±20%[*] | <±45% | <±60% | <±35% | <±55% | <±35%[*] |
| **NME** | **Goal** | <35% | <45% | <50%[*] | <45% | <40%[*] | <45%[*] |
| | **Criteria** | <45%[*] | <55% | <75%[*] | <55% | <60%[*] | <60%[*] |
| **FB** | **Goal** | <±15% | <±40% | <±20% | <±20% | <±25% | <±15% |
| | **Criteria** | <±25% | <±50% | <±75% | <±45% | <±45% | <55% |
| **FE** | **Goal** | <40% | <65% | <60% | <65% | <45% | <45% |
| | **Criteria** | <55% | <75% | <80% | <75% | <55% | <50% |

[a] Values with an asterisk in Table 2 indicate that our benchmarks are stricter than corresponding values in Emery et al. (2017)

[b] Shaded values indicate that less than 20 data points were available to develop the benchmarks.

**Report 2 by Referee #1**

Thanks for your revision, which addressed my concerns in part, but there are still some important issues have not been totally addressed. And I think, these issues could be critical for making this work valuable for other studies in future and therefore shaping this work be suitable for publishing in ACP.

We are grateful to the reviewer for taking time to carefully review the manuscript and give detailed and constructive comments, which has greatly helped to improve this paper. Below is our point-by-point response to each respective comment.

1) Authors include GEOS-Chem and conclude that GEOS-Chem's performance is less satisfied due to its relative coarse resolution leading to insufficient resolve details in interactions between emission and chemistry in a city-scale. Then, could authors provide more discussion in the manuscript to stress the necessary/advance of fine-res. models could be a promising trend of future air quality modelling studies to help improve our understanding.

**Response**: Thanks for the comment. We have added discussions regarding the application of GEOS-Chem in regional scale in the revised manuscript. GEOS-Chem is a global 3-D atmospheric chemistry model with state-of-science developments and large international user base. However, due to its coarse resolution, limitations exist when applying GEOS-Chem to simulations at regional or local scale. To tackle this issue, Lin et al. (2020) developed a new online regional atmospheric chemistry model - WRF-GC (v1.0), that integrates the WRF meteorology model and GEOS-Chem chemistry model. This new WRF-GC model has been successfully configured at finer resolution (27 km x 27 km) and applied to quantify the changes of NOx emissions due to COVID-19 for Eastern China (Zhang et al., 2020), illustrating the potential applications of GEOS-Chem at finer spatial scale. These discussions have been added to the revised manuscript.

**Revised manuscript (Page 8, Line 21-33):**

Fine resolution simulations have been conducted with the intention of improving model performance. With finer grid resolution, the spatial allocation of certain features in emission patterns is significantly improved, which is especially important for air quality simulations at local scale (Tan et al., 2015; Liu et al., 2020). Additionally, meteorological simulations could also be improved at finer resolution given more detailed land cover and structures in topography (Tao et al., 2020), which in turn improves the subsequent air quality simulations. Estimation of PM2.5 related health impacts are reported to be biased high/low at coarse spatial resolution (Li et al., 2017; Thompson and Selin, 2012). Lin et al. (2020) developed a new online regional atmospheric chemistry model - WRF-GC (v1.0), that integrates the WRF meteorology model and GEOS-Chem chemistry model. This new WRF-GC model has been configured with a spatial resolution of 27km and successfully applied to quantify the changes of NOx emissions due to COVID-19 for East China (Zhang et al., 2020), illustrating the potential applications of GEOS-Chem at finer spatial scale. However, not all fine resolution simulations lead to improved model performance, especially when the input data are not available with the same high resolution (Jiang and Yoo, 2018; Tao et al., 2020). Therefore, grid resolution should be determined depending on the purpose of the study and the availability of input data.

2) I do not object to your reply-02, however, reply-02 still did not address my question. Ozone is the central pollutant of photochemistry. And you do not talk a work about ozone in this manuscript, whose title highlight the "photochemical". Here is an example. A manuscript entitled "How deadly is COVID-19?", but it only talks about how to develop a vaccine, do not mention a word of global number of confirmed cases and death toll. Do you think this is a good title reflect the content presented? So, my point is, either title is not suitable or ozone need to be discussed.

**Response**: Thanks for the kind comment, we agree that ozone is a key air pollutant if we talk about "photochemistry". We changed the title from "photochemical grid model" to "Numerical air quality model". We use "Numerical" to make clear that we do not consider Lagrangian air quality models (for example, Hysplit, Calpuff) that have much simpler chemistry or none.

3) I do not agree that evaluation of meteorology is a "standalone scientific question". First, as suggest by author in the conclusion that performance of meteorology simulation can directly influence air quality simulation. Second, as replied by authors to Reviewer-02:

"Consequently, errors in inputs and algorithms present in this group of simulations are likely to be correlated (e.g., tendency for higher emissions correlated with more dispersive meteorological simulations), which will hinder reliable diagnosis of factors contributing to model error."

Exactly, a good air quality simulation can be achieved for wrong reasons when meteorological performance is poor, this will hind the reasons of uncertainty and hamper improvement in our understanding.

So, a convincing evaluation of air pollutants should always build on the top the evaluation of meteorology. I suggest to include meteorology in the work. If authors do insist to separate them, then at least, the meteorology evaluation work should publish first, and this work can cite it and build on the top of it.

Response: We totally agree with the reviewer that meteorological performance can influence the air quality simulation results to some extent and meteorological performance is an essential part of any air quality model evaluation. However, we decided not to include discussions on meteorological evaluation in this paper for the following reasons. First, not all of the 307 studies included in this study reported model performance results for their respective meteorological simulations, so the development of MPE benchmarks for meteorology would necessarily consider a different set of studies and lead to inconsistent meteorology-air quality performance connections. There are studies that performed evaluations for air quality simulations but did not mention meteorological evaluation and vice versa. Second, we see evaluations of meteorology and air quality are rather distinct issues, and it doesn't really matter which publication comes first. Indeed, the reviewer raised an important scientific question regarding how the meteorological MPE influences the air quality MPE, which is a very interesting topic and needs much more complex analysis. However, this is beyond the scope of the current study and we will possibly consider it in our following research. We have inserted explanations in the revised manuscript.

**Revised text (Page 5, Line 18-25):**
Meteorological data are needed to drive air quality simulations and the performance of meteorological modelling is a key source of uncertainty for air quality modelling performance. Meteorological data were mostly simulated by the Weather Research Forecasting (WRF) model (Skamarock et al., 2005) in our compiled studies; the Fifth Generation Penn State/NCAR

Mesoscale Model (MM5) (Grell et al., 1994) and the Regional Atmospheric Modelling system (RAMS) were used in a few studies. Model performance of meteorological results should be evaluated in addition to air quality simulation results. However, several studies did not report any results with respect to their meteorological simulations. The performance of meteorological results used to drive air quality simulations and how it could affect the air quality simulations is beyond the scope of the current work and will need to be discussed as a future work.

4) in reply-04, authors state "Our results indicate that the top three factors involve emission inventory, grid resolution, and boundary conditions, while the choice of model and source of meteorology are least important. This is a very preliminary analysis and thus we have decided to include these discussions in the supporting materials.".

I think this is the most important part of the present work, it could be the unique contribution to the modelling community and hence a unique value of this work. We do want to have detailed discussions and promote the discussions to main text. Even though there are limitations, we could still have discussions of these limitations (all studies have limitations) and the uncertainties, and provide valuable advices for future studies to overcome these limitations. Put this part in SI really under value the present work.

**Response**: Thanks for the positive comment. We agree that moving this part to the main text could improve the depth of this study. As suggested by the reviewer, we moved this part to the main work.

Revised manuscript:

**2.4 Feature importance based on Random Forest**

Random Forest is a machine learning method suitable for classification and regression (Liu et al., 2012). It is a collection of a series of decision trees and each tree is generated from a bootstrap sample. Both continuous and categorical input variables are allowed. It can provide the order of feature importance (FI) so that we can determine and rank which parameter choices most influence the simulation results.

We reviewed the model configurations for studies that reported correlation coefficient, IOA, MB, NMB, mean error (ME), normalized mean error (NME), fractional bias (FB), and fraction error (FE) for $PM_{2.5}$ (a total of 176 studies). Model configurations include the meteorological data that are used to drive air quality simulations (e.g. from WRF, MM5, or GEOS), the emission inventory (e.g. public available dataset vs. locally developed), gas-phase chemistry (for example, carbon bond vs. Statewide Air Pollution Research Center (SAPRC)), aerosol chemistry (including inorganic aqueous chemistry, inorganic gas-particle partitioning, organic gas-particle partitioning and oxidation), boundary conditions (e.g. model default values vs. results generated from global model), grid resolution and the temporal resolution (Table S7). We ignored the study region and period for FI selection because these two options are more restricted by the user's specific needs and focus (i.e., more subjective/uncontrollable and less objective/controllable). We ranked each statistical metric from good to poor performance. For example, values of R and IOA that are close to 1 represent good performance and values close to 0 represent poor performance. For MB and NMB, we used absolute values so that deviations from zero represent the performance level. These results were classified into three tiers with breaks at 33% and 67% of the ranked values so that each tier includes the top one third, the middle one third, and the bottom one third of the reported performance results. The random forest model was performed using the 'sklearn' module in Python to obtain the FI metric.

**3.3 Recommended metrics and benchmarks**

As mentioned earlier, AQM applications involve numerous driving inputs as well as diverse model configurations, which lead to an abundant database from which to assess their relative influences on model performance. The similarities between the benchmarks derived in this study and Emery's study suggest that important model input data (e.g. emission inventories) have comparable accuracy for China and North America and model formulations (e.g. algorithms such as chemistry, deposition, transport) seem to be equally applicable to China and North America. In additional to the need for model performance benchmarks, there also is a need for more studies that quantify contributions to model uncertainty, such as the recent study by Dunker et al. (2020), which quantifies contributions of chemistry, boundary concentrations, deposition and emissions to uncertainty in simulated ozone results. In this study, we applied the Random Forest method for pattern recognition to identify and rank model attributes (inputs, grid resolutions, etc.) that have important influences on $PM_{2.5}$ model performance. The choice of emission inventory is shown to affect the model performances most, followed by grid resolution, aerosol and gas chemistry (Figure 11). Meteorological input and the choice of model itself is of least importance.

---

## Author Response (AR3)

Reviewer's comment:

Thanks for the revision. It has addressed most of my concerns. Would authors also add some comments on the relationship between meteorology uncertainty and air pollutants uncertainty. A poor meteorology simulation, air quality simulations could achieve a good evaluation due to wrong reason, especially for evaluations based on monthly mean. And this could hidden the important sources of uncertainty in the model, may not be good for model improvements. I think after the revision, this work could be considered for publishing in ACP.

Response: Thanks for this comment. We have added some discussions with respect to this comment in the revised manuscript.

In revised manuscript (Page 11, Line 17-23):
"Meteorological information is an essential input to each air quality simulation (along with emissions, boundary concentrations, etc.) and uncertainties in the meteorology will inevitably influence the air quality simulation to some degree. Indeed, meteorological errors could be offset by errors in other model inputs thus resulting in good air quality performance evaluation results for the wrong reasons (Reynolds et al., 1996). For example, the effect of low-biased wind speed could be offset by low-biased emissions, or vice versa, producing simulated air quality in agreement with observations but incorrect response of air quality to emission changes. Therefore, evaluating the meteorological model performance is as important as air quality model performance evaluation."

References:
Reynolds, S., Michaels, H., Roth, P., Tesche, T.W., McNally, D., Gardner, L. and Yarwood, G., 1996. Alternative base cases in photochemical modeling: their construction, role, and value. Atmospheric Environment, 30(12), pp.1977-1988.